# Persistent epigenetic memory of SARS-CoV-2 mRNA vaccination in monocyte-derived macrophages

Alexander Simonis [1,2,3,16], Sebastian J Theobald[1,2,3,16], Anna E Koch [2], Ram Mummadavarapu [4], Julie M Mudler [1,2], Andromachi Pouikli[4], Ulrike Göbel[5], Richard Acton [5,6], Sandra Winter[1,2], Alexandra Albus[1,2], Dmitriy Holzmann[1,2], Marie-Christine Albert[5,7], Michael Hallek[1,2], Henning Walczak [5,7,8], Thomas Ulas[9,10,11], Manuel Koch [7,12], Peter Tessarz [4,5,13], Robert Hänsel-Hertsch [2,5,14,15,16 ✉] & Jan Rybniker [1,2,3,16 ✉]

## Abstract

**Immune memory plays a critical role in the development of durable antimicrobial immune responses. How precisely mRNA vaccines train innate immune cells to shape protective host defense mechanisms remains unknown. Here we show that SARS-CoV-2 mRNA vaccination significantly establishes histone H3 lysine 27 acetylation (H3K27ac) at promoters of human monocyte-derived macrophages, suggesting epigenetic memory. However, we found that two consecutive vaccinations were required for the persistence of H3K27ac, which matched with pro-inflammatory innate immune-associated transcriptional changes and antigen-mediated cytokine secretion. H3K27ac at promoter regions were preserved for six months and a single mRNA booster vaccine potently restored their levels and release of macrophage-derived cytokines. Interestingly, we found that H3K27ac at promoters is enriched for G-quadruplex DNA secondary structure-forming sequences in macrophage-derived nucleosome-depleted regions, linking epigenetic memory to nucleic acid structure. Collectively, these findings reveal that mRNA vaccines induce a highly dynamic and persistent training of innate immune cells enabling a sustained pro-inflammatory immune response.**

**Keywords** Epigenetic Memory; Trained Innate Immunity; SARS-Cov-2 mRNA Vaccination; H3K27ac; G-quadruplex
**Subject Categories** Chromatin, Transcription & Genomics; Immunology; Microbiology, Virology & Host Pathogen Interaction

## Introduction

SARS-CoV-2 mRNA vaccines have been essential in controlling the coronavirus disease 2019 (COVID-19) pandemic (Baden et al, 2021; Polack et al, 2020; Yang et al, 2023). How exactly these novel vaccine constructs, which drive mRNA-dependent expression of the SARS-CoV-2 spike protein (SP), activate different layers of the innate and adaptive immune system remains not fully understood (Verbeke et al, 2022). There is strong evidence that pathogen-specific immune responses require durable activation of innate immune cells, such as macrophages or dendritic cells via adjuvants and pathogen-associated molecular patterns (PAMP). The quality, quantity and extent of innate immune cell activation dictates long-lived memory responses of adaptive immune cells such as B and T cells (Fitzgerald and Kagan, 2020; Iwasaki and Medzhitov, 2010; Sonnenberg and Hepworth, 2019).

Recently, we were able to show that both SARS-CoV-2 infection and mRNA vaccination prime human monocyte-derived macrophages for potent secretion of pro-inflammatory cytokines following restimulation with the SARS-CoV-2 SP ex vivo (Theobald et al, 2021; Theobald et al, 2022). In our studies, we focused on the key cytokine interleukin-1β (IL-1β), which we linked to macrophage-driven activation of effector memory T cells using autologous co-culture experiments (Theobald et al, 2022). Secretion of IL-1β depends on activation of the highly regulated NOD-, LRR-, and pyrin domain-containing protein 3 (NLRP3) inflammasome in macrophages (Swanson et al, 2019). In both vaccines and natural infections, inflammasome-derived IL-1β and associated receptors play a key role in transmitting stimulatory signals between innate and adaptive immune cells (Tahtinen et al, 2022; Van Den Eeckhout et al, 2020). Surprisingly, we found SP-dependent

[1]Department I of Internal Medicine, Faculty of Medicine and University Hospital Cologne, University of Cologne, Cologne 50937, Germany. [2]Center for Molecular Medicine Cologne (CMMC), Faculty of Medicine and University Hospital Cologne, University of Cologne, Cologne 50931, Germany. [3]German Center for Infection Research (DZIF), Partner Site Bonn-Cologne, Cologne, Germany. [4]Max Planck Research Group "Chromatin and Ageing", Max Planck Institute for Biology of Ageing, Joseph-Stelzmann-Str. 9b, Cologne 50931, Germany. [5]Excellence Cluster on Cellular Stress Responses in Aging-Associated Diseases (CECAD), University of Cologne, Cologne, Germany. [6]Babraham Institute, Cambridge, UK. [7]Institute of Biochemistry I, Faculty of Medicine, University Hospital Cologne, University of Cologne, Cologne, Germany. [8]Center for Cell Death, Cancer and Inflammation, UCL Cancer Institute, University College London, London, United Kingdom. [9]Systems Medicine, German Center for Neurodegenerative Diseases (DZNE), University of Bonn, Bonn, Germany. [10]PRECISE Plattform for Single Cell Genomics and Epigenomics, DZNE, University of Bonn, Bonn and West German Genome Center, Bonn, Germany. [11]Genomics and Immunoregulation, Life & Medical Sciences (LIMES) Institute, University of Bonn, Bonn, Germany. [12]Institute for Dental Research and Oral Musculoskeletal Biology, Center for Dental, Oral and Maxillofacial Medicine (central facilities), Medical Faculty and University of Cologne, Cologne, Germany. [13]Department of Human Biology, Radboud Institute for Molecular Life Sciences, Faculty of Science, Radboud University, Nijmegen, The Netherlands. [14]Department of Translational Genomics, Faculty of Medicine and University Hospital Cologne, University of Cologne, Cologne, Germany. [15]Institute of Human Genetics, University Hospital Cologne, Cologne, Germany. [16]These authors contributed equally: Alexander Simonis, Sebastian J Theobald, Robert Hänsel-Hertsch, Jan Rybniker. ✉E-mail: robert.haensel-hertsch@uni-koeln.de; jan.rybniker@uk-koeln.de

maturation of IL-1β to be highly selective with no or very little IL-1β secreted in stimulated macrophages of non-vaccinated or SARS-CoV-2 non-infected individuals. Furthermore, vaccination-induced macrophage priming could be enhanced with repetitive antigen exposure following the second SARS-CoV-2 mRNA vaccination (Theobald et al, 2022). After a transient decline of macrophage responsiveness, we observed potent secretion of IL-1β from macrophages after the third (second booster) vaccination, which was applied after six months. Consequently, the data suggest that monocyte-derived macrophages maintain durable alterations for several months following SARS-CoV-2 mRNA vaccination, despite the fact that circulating monocytes have a lifespan that typically does not exceed seven days (Whitelaw, 1966). Recent studies primarily performed in mice, found that innate immune cells maintain long-lived epigenetic memory following exposure to pathogen-derived antigens, allowing for rapid and potent responses towards subsequent challenges. This innate immune memory, also termed trained innate immunity, has been linked to long-lived chromatin modifications in hematopoietic progenitor cells, which serve as epigenetic marks allowing mature innate immune cells to rapidly adjust their transcriptional profile in response to pathogen-derived antigens (Netea et al, 2016).

Notably, epigenetic remodeling extends beyond chromatin modifications and has also been linked to alternative DNA secondary structures. Human DNA can adopt G-quadruplex (G4) structures in repetitive elements, such as telomeres, as well as in nucleosome-depleted regions, particularly at highly transcribed genes (Biffi et al, 2013; Hansel-Hertsch et al, 2016; Hansel-Hertsch et al, 2020). G4 DNA has been associated with epigenetic remodeling and the regulation of innate immune responses (Guilbaud et al, 2017; Makowski et al, 2018; Miglietta et al, 2021).

Epigenetic modifications have been observed following vaccination using Bacillus Calmette-Guérin (BCG), a live-attenuated vaccine derived from *Mycobacterium bovis*, which is primarily used for tuberculosis prevention. This vaccine has been shown to induce epigenetic changes and enhance innate immune memory (Sun et al, 2024). It is postulated that this memory response may also confer protection against non-related infectious diseases (Kaufmann et al, 2018; Kleinnijenhuis et al, 2012). Similar examples were provided for other live-attenuated vaccines such as vaccinia virus or the influenza vaccine (Sanchez-Ramon et al, 2018). Whether inactivated vaccines or mRNA-based vaccines, which are believed to revolutionize future vaccination strategies, lead to long-term alterations of innate immune cells with corresponding epigenetic alterations is unknown. Here we show that SARS-CoV-2 mRNA vaccines induce persistent epigenetic and transcriptomic changes in monocyte-derived macrophages leading to an increased reactivity to a range of SARS-CoV-2 related and non-related PAMPs.

# Results

## Two SARS-CoV-2 mRNA vaccinations are required to prime a broad pro-inflammatory immune response in macrophages

In previous studies, we could demonstrate that SARS-CoV-2 mRNA vaccination and SARS-CoV-2 infection can lead to significant alterations in monocytes and monocyte-derived macrophages that last beyond the

cells' typical lifespan (Theobald et al, 2021; Theobald et al, 2022). These phenotypic studies primarily relied on restimulation of macrophages with the affinity-purified SARS-CoV-2 spike protein (SP) ex vivo. To investigate in detail whether SARS-CoV-2 mRNA vaccination induces a comprehensive and non-specific reprogramming of monocyte-derived macrophages, which is a key characteristic of innate immune memory, we performed extensive restimulation assays with diverse PAMPs and danger-associated molecular patterns (DAMP) targeting multiple pattern recognition receptors (Fig. 1A,B). Exploiting our large longitudinal cohort of SARS-CoV-2 mRNA-vaccinated individuals, we first evaluated the magnitude of IL-1β secretion in macrophages incubated with single-stranded RNA (ssRNA) (Toll-like receptor (TLR) 7 and TLR8 agonist), zymosan (TLR2 and Dectin-1 agonist), and Pam3CSK4 (TLR2/TLR1 agonist) (Dataset EV1). By stimulating monocyte-derived macrophages from vaccinated individuals before vaccination (t0), 2 weeks after the first (t1) and second (t2) vaccination as well as 10 weeks after the second vaccination (t3), we observed an increase of secreted IL-1β compared to t0 (Fig. 1C–E). Of note, for all receptor ligands used, statistically significant IL-1β levels were only observed 2 weeks after the second vaccination (t2) indicating that a prime-boost regimen is required for substantial activation and reprogramming of macrophages. For cells stimulated with ssRNA, reactivity remained preserved for up to 10 weeks after the second vaccination (Fig. 1C).

IL-1β secretion induced by the different receptor ligands which were used for ex vivo stimulation is associated with activation of inflammasomes and pyroptosis, a highly pro-inflammatory form of programmed cell death characterized by membrane rupture (Man et al, 2017). To confirm that additional pro-inflammatory pathways are primed in monocyte-derived macrophages of vaccinated individuals, we performed multiplex cytokine analyses of SP-stimulated macrophages showing that several cytokines and chemokines (TNF-α, IL-36, CCL3, CCL20, CCL4, CXCL1) were significantly elevated in supernatants of macrophages at t2 compared to those isolated from unvaccinated individuals (Figs. 1F–I and EV1A,B). Other analytes, such as CCL7, CXCL13, and CXCL10 were not affected by SP stimulation in macrophages at t0 and t2 (Fig. EV1C–E).

Finally, to demonstrate the essential role of the master regulator nuclear factor 'kappa-light-chain-enhancer' of activated B cells (NF-κB) in cytokine secretion of stimulated macrophages, we treated t2 macrophages with KINK-1 (kinase inhibitor of NF-κB) and quantified IL-1β in the cell supernatants. MCC950, a selective NLRP3 inflammasome inhibitor was used as a control substance. Both inhibitors abrogated IL-1β secretion in response to all three receptor agonizts significantly (Fig. EV1F–H).

Taken together, these findings confirm that SARS-CoV-2 mRNA vaccination enhances the responsiveness of in vivo-primed macrophages to various immunological triggers, which is mediated by NF-κB and inflammasome signaling. In our cohort, two vaccinations were required for the release of significant amounts of cytokines in stimulated cells suggesting the induction of a memory response by the first priming vaccine in monocyte-derived macrophages.

## Cytokine responses match with global transcriptomic changes in monocyte-derived macrophages of vaccinated individuals

Intrigued by our observation that SARS-CoV-2 mRNA vaccination broadly enhances the cytokine response to different PAMPs, we

   

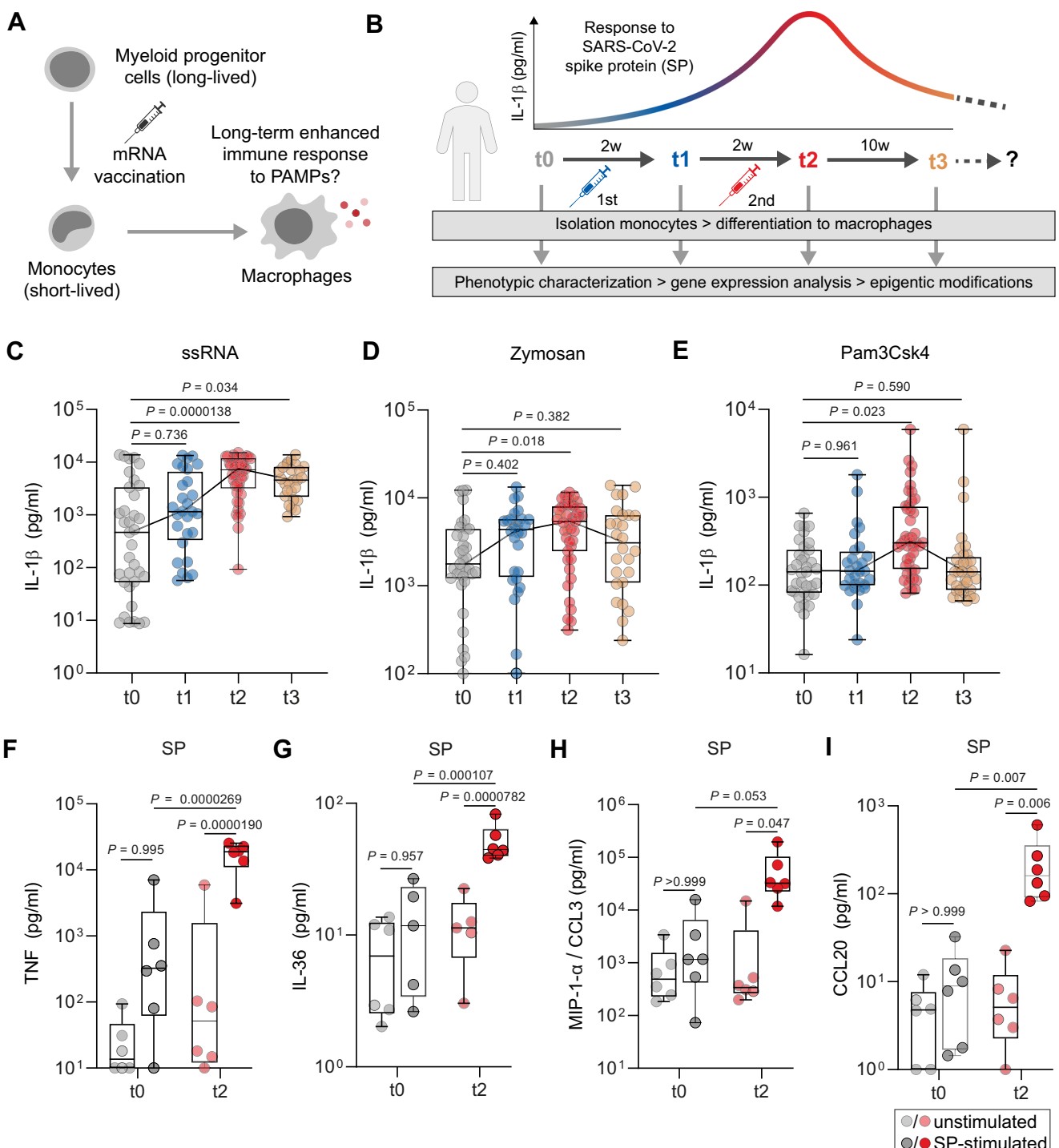

performed RNA-sequencing analyses of SP-stimulated and unstimulated monocyte-derived macrophages at t0 and t2 to comprehensively quantify the impact of vaccination on a genome-wide transcriptional level. Here, we observed substantial differences in gene expression patterns between the different groups. SP-stimulation of monocyte-derived macrophages from unvaccinated individuals (t0) induced only moderate changes in gene expression levels compared to unstimulated t0 cells (Fig. 2A; Dataset EV2). In contrast, SP stimulation of cells isolated at t2 led to a tremendous

shift of the transcriptome indicating a priming effect of the vaccination of subsequent gene expression upon SP stimulation (Fig. 2B; Dataset EV3). When comparing the transcriptome of SP-stimulated cells of both groups (unvaccinated versus vaccinated), the two groups were clearly distinguishable based on their gene expression profiles (Fig. 2C). Gene expression analysis in macrophages following SP stimulation, compared to unstimulated cells, revealed 2519 genes that were differentially expressed (defined by a log2 fold change of ±1 and an adjusted *P* value < 0.05) exclusively in

◀ **Figure 1. Cytokine release in macrophages following SARS-CoV-2 mRNA vaccination.**

(A) Graphical illustration of the working hypothesis. (B) Study design: Blood samples from healthy donors were collected in a longitudinal manner for monocyte isolation, followed by differentiation to macrophages by cultivation with M-CSF and used for downstream applications including ex vivo stimulation experiments, RNA-seq- and CUT&RUN analyses. Samples were collected prior (t0) and 2 weeks after first vaccination (t1), 2 weeks after second vaccination (t2), which was applied 4 weeks after the first vaccination and 10 weeks after the second vaccination (t3). The upper graph illustrates IL-1β secretion at different time points upon stimulation with SARS-CoV-2 spike protein (SP) as shown in previous studies. (C) Monocytes were isolated by CD14$^+$ selection from peripheral blood mononuclear cells (PBMCs). Cells were seeded and incubated in the presence of M-CSF for 5 days. Differentiated macrophages were stimulated with ssRNA (t0: $n = 35$, t1 $= 28$; t2: $n = 42$; t3: $n = 26$), Zymosan (t0: $n = 35$, t1: $n = 28$; t2: $n = 43$; t3: $n = 26$) (D) or Pam$_3$Csk$_4$ (t0: $n = 35$, t1: $n = 28$; t2: $n = 44$; t3: $n = 33$) (E) for 4 h. IL-1β secretion was quantified by ELISA. For statistical analysis, one-way ANOVA with Dunnett's multiple comparison test comparing t1–t3 to t0 was used. (F) Monocyte-derived macrophages from unvaccinated (t0) (gray) and vaccinated individuals (t2) (red) were generated, stimulated with SP (t0: dark gray dots; t2: dark red dots) or left unstimulated (t0: light gray dots; t2: light red dots) as described in (C). Concentrations of TNF, IL-36 (G), MIP-1-α (CCL3) (H), and CCL20 (I), were measured by multiplex analyses. For statistical analysis, two-way ANOVA with Sidak's multiple comparison analysis was used. Box plots indicate the median, the upper and lower quartile and the minimum and maximum values. Shown data points represent the technical mean of an independent experiment. *P* values less than 0.05 were considered statistically significant. Source data are available online for this figure.

macrophages derived from vaccinated individuals. In addition, 268 differentially expressed genes (DEGs) were shared between both groups, while only 50 DEGs were specific to macrophages from unvaccinated individuals (Fig. 2D).

Furthermore, gene enrichment analyses revealed a strong upregulation of genes associated with the innate immune response, cytokine signaling and the defense response to viruses in macrophages from vaccinated individuals stimulated with SP at t2, compared to those from unvaccinated individuals (Fig. 2E,F; Datasets EV4 and EV5). These observations further support our hypothesis that mRNA vaccination induces a priming effect, enhancing transcriptional reactivity upon exposure to pathogen-associated stimuli ex vivo.

## Vaccination instructs persistent H3K27ac level in gene promoter

Epigenomic changes of short-lived myeloid cells are necessary to enhance innate immune signaling upon restimulation with PAMPs and DAMPs as observed in our study (Kleinnijenhuis et al, 2012; Mitroulis et al, 2018; Quintin et al, 2012). The responsiveness of trained innate immune cells to time-delayed stimulation via epigenetic alterations represents one of the hallmarks of innate immune memory (Netea et al, 2020; Sherwood et al, 2022). These alterations require chromatin remodeling mediated by histone modifications such as acetylation or methylation which result in transcriptional activation or repression (Foster et al, 2007; Saeed et al, 2014). To examine these effects in individuals having received SARS-CoV-2 mRNA vaccines, we focused on histone 3 lysine 27 acetylation (H3K27ac) as a marker for active enhancers and promoters in monocyte-derived macrophages (Wang et al, 2008). To determine the level and persistence of H3K27ac in short-lived macrophages, we extended our study and included two additional time points prior to and after the third (second booster) SARS-CoV-2 mRNA vaccination (t4 and t5). This vaccine was applied 24 weeks (6 months) after administration of the first two vaccines (Dataset EV1).

To determine the level and genome-wide locations of H3K27ac at the five post vaccination time points, t1 to t5, we performed *cleavage under targets and release using nuclease* (CUT&RUN). Importantly, and to assess whether vaccination leads to an increase and persistence in H3K27ac at genes, we also mapped H3K27ac prior to vaccination (t0) and normalized H3K27ac sequencing read coverages of post vaccination time points (t1 to t5) relative to t0. Interestingly, this approach revealed a highly dynamic landscape of

H3K27ac ratio changes at different time points. An example genome browser view highlights a moderate increase after the first vaccination (t1), followed by a substantial increase after the second vaccination (t2), then a declining trend from t2 to t3 toward t4, and finally a marked increase after the third vaccination (t5) (Fig. 3A). To systemically address H3K27ac alterations across time points, we first globally assessed the levels of H3K27ac enriched genomic regions by peak calling and associated these H3K27ac peaks by their closest distance to annotated genes. Active enhancers are found in gene bodies and in distal regulatory regions. To assess whether gene-associated peaks gain H3K27ac in enhancers or promoters in response to vaccination, we categorized H3K27ac peaks as (a) promoter, 1 kb upstream to 250 bp downstream of transcription start sites (TSS), (b) gene body, 251 bp downstream of TSS to transcription end sites (TES), and (c) distal, more than 1 kb to 100 kb away from TSS and TES (Fulco et al, 2019). Using this classification, we identified an overall expected short-term increase in H3K27ac levels after all vaccination time points (t1, t2, t5) in promoters and enhancer regions, including gene bodies and distal regions (Fig. 3B–D; Dataset EV6). Strikingly, at t4, 34 weeks after the second vaccination (t2), we found persistent, not declining, H3K27ac level at gene promoters but not in gene bodies and distal regions of macrophages obtained from vaccinated individuals as opposed to those who had not received the vaccine (Figs. 3E and EV2A). This strongly suggests that promoters acquire an epigenetic memory upon vaccination. Pioneering transcription factors can actively or passively establish accessible chromatin in monocytes that drive gene expression programs and are transmitted as epigenetic memory to monocyte-derived macrophages (Saeed et al, 2014; Zaret and Carroll, 2011). Endogenous G4 DNA secondary structures are prevalent in highly transcribed, accessible promoters, which are hotspot targets for transcription factors (Hansel-Hertsch et al, 2016; Hansel-Hertsch et al, 2020; Spiegel et al, 2021). Perturbation of G4 DNA structures affect nucleosome density, opening the possibility that they could support persistent accessibility of promoters for the transcriptional machinery (Esain-Garcia et al, 2024; Esnault et al, 2023). To address whether the promoters marked by persistent H3K27ac contain the potential to adopt G4 secondary structure, we considered experimentally validated G4-sequences in the human genome that overlap with annotated accessible genomic regions of macrophages (Marsico et al, 2019; Zou et al, 2022). Analysis of G4 sequence coverage in H3K27ac peaks revealed a substantial increase in G4 DNA-forming sequences in promoters but not in gene bodies and distal regions, also relative to randomized expectation (see methods) (Fig. 3F). Together, these results suggest that H3K27ac persist for

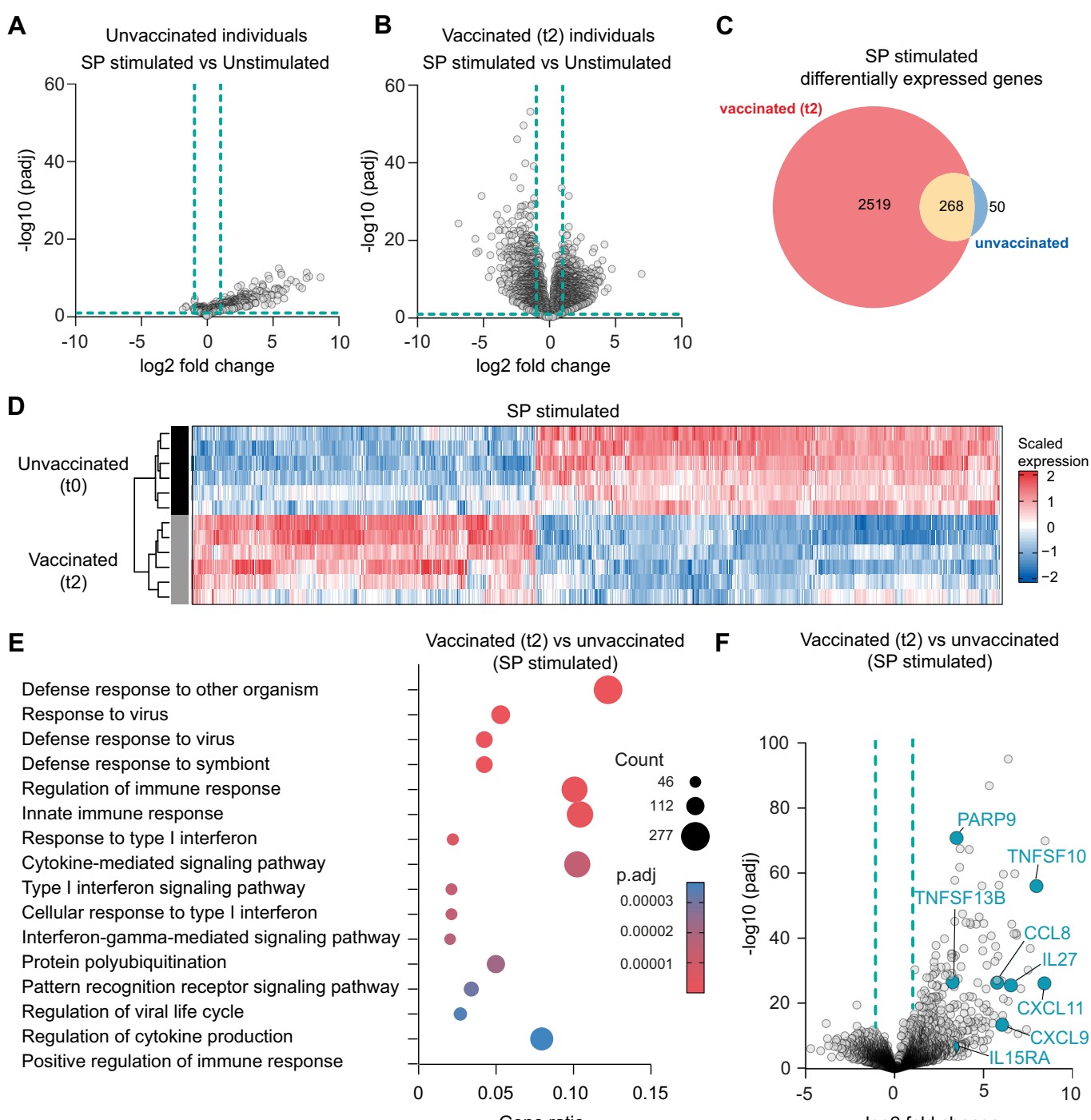

several months at promoters following vaccination and that this is linked to G4 structural potential in accessible chromatin of macrophages.

## Epigenetic H3K27ac memory is established in promoters of vaccine-responsive immune genes

To identify gene-specific clusters that acquire similar gain and loss of H3K27ac throughout the history of vaccination (t1 to t5 relative to t0), we performed unsupervised hierarchical clustering of H3K27ac at

promoters, gene bodies and distal regions across the time points with respect to unvaccinated (t0) individuals (Fig. 4A–C). Heatmaps displaying these data clearly revealed one dominant gene cluster (cluster 1, ~8000 genes) related to H3K27ac in promoters (Fig. EV2B), while a higher diversity of different gene clusters was observed in gene bodies and distal regions (Fig. 4A–D). Strikingly, we found that persistent H3K27ac levels were significantly higher in the genes of cluster 1 at t4 compared to t1 macrophages. However, this was only the case when peaks were located in promoters of cluster 1, not in gene bodies or distal regions. This analysis indicates that not all but

**Figure 2. Trancriptomic changes in macrophages determined with RNA-sequencing.**

(A) Volcano plot showing differentially expressed genes (gray dots) in SP-stimulated macrophages ($n = 6$) compared to unstimulated macrophages ($n = 6$) at t0 for unvaccinated (**A**) and vaccinated (t2) (**B**) individuals. Negative log10 adjusted $P$ values are plotted against the log2 fold change. Downregulated genes are depicted with negative log2 fold-change values on the left side of the plot, whereas upregulated genes are represented by positive log2 fold-change values on the right side. Dotted lines indicate log2 fold-change ±1 and −log10 adjusted $P$ values of 1. Genes were considered differentially expressed if they showed a log2 (fold change) > 1 and were below an FDR of 0.05. (**C**) Venn diagram showing number of differentially expressed genes (DEGs) in monocyte-derived macrophages of unvaccinated (blue) ($n = 6$) and vaccinated (red) ($n = 6$) individuals upon SP stimulation compared to unstimulated cells. The red color represents the number of DEGs unique to vaccinated individuals, while the blue color indicates DEGs unique to unvaccinated individuals. The yellow color highlights the overlapping DEGs between the two groups. Circle sizes of the venn diagram correspond to the number of genes. (**D**) Heatmap indicating DEG patterns comparing SP stimulation of monocyte-derived macrophages from unvaccinated ($n = 6$) or vaccinated (t2) ($n = 6$) individuals. Gene expression levels (log2 normalized expression values) are color-coded as indicated. (**E**) Gene ontology (GO) enrichment analysis, based on DEGs of monocyte-derived macrophages from vaccinated (red) ($n = 6$) individuals compared to unvaccinated individuals ($n = 6$) after stimulation with SP. $P$ adjusted values are indicated (color code) and sizes of the circles represent number of DEGs (count). Gene ratio ($x$ axis) indicate the percentage of the number of genes present in this GO term over the total number of genes in this category. Data are shown for the top 15 biological processes ranked by adjusted $P$ values. The $P$ value cutoff was 0.05 by permutation, (using the default values in clusterProfiler), for genes rank ordered by fold change (log2). (**F**) Volcano plot showing DEGs in SP-stimulated macrophages from vaccinated individuals ($n = 6$) at t2 compared to stimulated cells from unvaccinated individuals ($n = 6$). Negative log10 adjusted $P$ values are plotted against the log2 fold change. Selected genes are labeled. Genes were considered differentially expressed if they showed a fold change (log2) > 1 and were below an FDR of 0.05.

certain promoters display a persistent level of H3K27ac after two vaccinations and a duration of 34 weeks (Fig. 4E–G).

Next, we assessed whether any of the gene clusters from all categories (promoter, gene body, distal) would be associated to gene ontology (GO) terms associated with immune functions. Systematic investigation revealed, indeed, that 70% of all GO terms in gene cluster 1 were related to the human immune response (e.g., positive regulation of leukocyte activation; leukocyte activation involved in immune response; lymphocyte activation and differentiation) (Dataset EV7). Interestingly, we identified a significant overlap of H3K27ac peaks in promoters of immune-associated genes ($n = 333$), which are a subset of cluster 1, with differentially expressed genes of SP-stimulated macrophages derived from vaccinated individuals (Fig. EV2C,D). A similar overlap was not observed for randomly picked promoters, indicating an association between epigenetic and transcriptomic datasets (Fig. EV2C,D).

Importantly, these immune-associated genes such as *IL1B*, *IL-18*, and *SYK*, have previously been linked with the innate immune response to SARS-CoV-2 infection and mRNA vaccination (Cheong et al, 2023; Theobald et al, 2021; Theobald et al, 2022) (Fig. 5A). Furthermore, persistent H3K27ac at promoters of these genes displayed increased coverage of G4 DNA sequences from macrophage-specific nucleosome-depleted regions, however, not in randomized annotated promoters, suggesting a macrophage-specific enrichment of G4 DNA sequences at immune promoters relative to their random distribution in promoters (Fig. 5B). Zooming into the respective genetic regions coding for IL-1β, IL-18, SYK, NOD2, and several c-type lectin (CLEC) family members which are linked to SYK and inflammasome signaling, we found that the H3K27ac peaks associated with these genes were more pronounced at time points t2, t3, and t5 compared to t1 and t4 (Figs. 5C, EV2E, and EV3A,B).

Finally, to also phenotypically confirm persistent immune memory in macrophages, we measured IL-1β secretion at t4 and t5 in response to PAMPs. As expected, we detected significantly increased levels of IL-1β in supernatants of simulated t5 macrophages when compared to t4 macrophages, linking increased cytokine secretion to H3K27ac levels and enhanced immune responsiveness (Fig. 5D). Notably, IL-1β release was substantially higher at t5 relative to t4 than at t1 relative to t0 (zymosan stimulation: median fold change of 16.38 [t5/t4] vs. 1.63 [t1/t0]; Pam3CSK4 stimulation: median fold change of 66.90 [t5/t4] vs. 0.92 [t1/t0]) again indicating a memory effect that persists over several months (Fig. EV4). The lower effect of ssRNA stimulation on IL-1β secretion (median fold change of 2.59 [t5/t4] vs. 6.28 [t1/t0]) might be explained by sustained long-term responsiveness to ssRNA after two vaccinations (t3) which was not observed to this extent for zymosan or Pam3CSK4 (Fig. 5D).

In summary, we demonstrate that SARS-CoV-2 vaccination induces long-term epigenetic modifications in H3K27ac at promoter regions, resulting in enhanced gene expression and innate immune responses to unrelated PAMPs in macrophages, thereby suggesting the induction of trained innate immunity.

# Discussion

In our study, we exploit the opportunity to comprehensively investigate the cytokine response as well as transcriptional and epigenetic alterations of monocyte-derived macrophages in an immunologically naïve population receiving mRNA vaccinations, which represent novel and highly promising vaccine constructs for a multitude of clinical applications.

We were able to demonstrate that SARS-CoV-2 mRNA vaccination establishes extensive and persistent H3K27ac at promoters of short-lived macrophages. However, a prime-boost vaccination regimen was required to achieve significant levels of epigenetic reprogramming lasting for several months after application of the second vaccine. The priming vaccine alone had little impact on this epigenetic mark associated with an altered immune response. Importantly, the dynamic epigenetic landscape we observed could be linked to the capability of macrophages to secrete pro-inflammatory cytokines upon stimulation with a series of pathogen- or vaccine-derived innate immune triggers. Thus, our observations are well in line with the hallmarks of innate immune memory or trained innate immunity defined as the long-term functional reprogramming of mature myeloid cells (Netea et al, 2020). Months-long innate immune memory of short-lived cells is achieved via epigenetic reprogramming of bone marrow progenitor cells and blood-derived monocytes (Netea et al, 2020). A key epigenetic enhancer mark in these cells is the acquisition of histone 3 lysine 27 acetylation (H3K27ac), which we observed to increase at promoters following mRNA vaccination. Indeed, this non-canonical long-term establishment of H3K27ac memory at promoters is surprising given its canonical role in distal regions, such as enhancers and gene bodies, but may be explained by their notable enrichment of G4 DNA sequences that act as a biophysical counterforce to prevent nucleosome-mediated DNA condensation

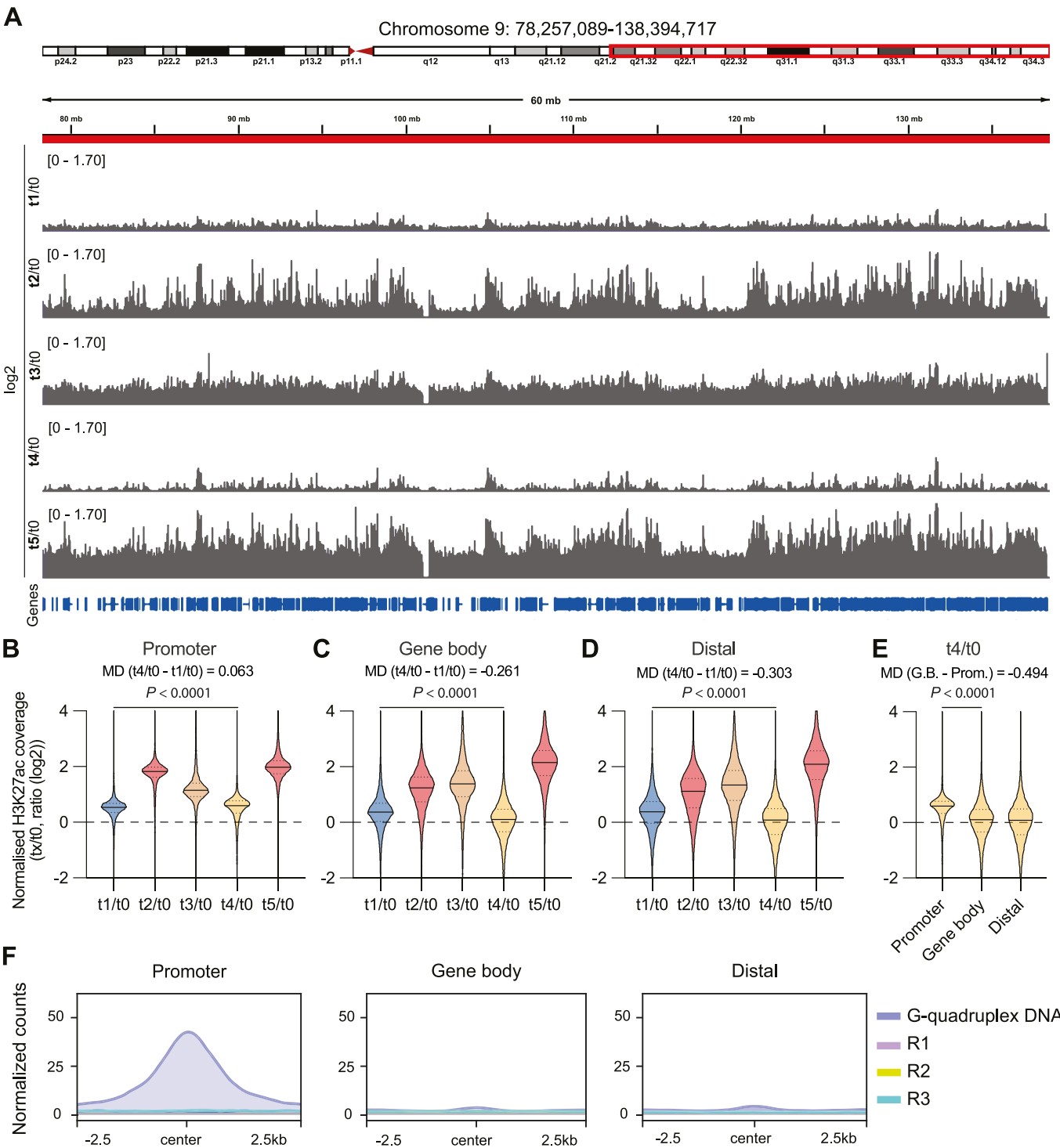

**Figure 3. Vaccination induces an increase and sustained persistence of H3K27ac at gene promoters.**

(A) Genome browser view of 60 mb of chromosome 9, displaying ratio (log2) of tx over t0 mean H3K27ac coverage across the time points. (B) Distribution of H3K27ac coverage ratios (log2) in peaks of promoters ($n = 17,026$) (1 kb upstream to 250 bp downstream of transcription start sites [TSS]) (B), gene bodies ($n = 19,583$) (251 bp downstream of TSS to transcription end sites [TES]) (C) and distal regions ($n = 19,315$) (1 kb to 100 kb away from TSS and TES) (D) across tx relative to t0. (B–D) All distribution comparisons are significantly different ($P < 0.001$) (see Dataset EV6). Median difference (MD) between the t4/t0 and t1/t0 ratio is indicated for all three genomic regions. (E) H3K27ac coverage ratios (log2) in peaks of promoters is shown for t4 compared to t0 for promoters, gene bodies and distal genes. MD is indicated between gene bodies (G.B.) and promoters. All distribution comparisons are significantly different ($P < 0.0001$). $P$ values were calculated using RM One-way ANOVA with Tukey's multiple comparison test. $P$ values less than 0.05 were considered statistically significant. (F) G-quadruplex DNA coverage in H3K27ac peaks located in annotated macrophage-specific accessible chromatin in promoters, gene bodies, distal regions. Source data are available online for this figure.

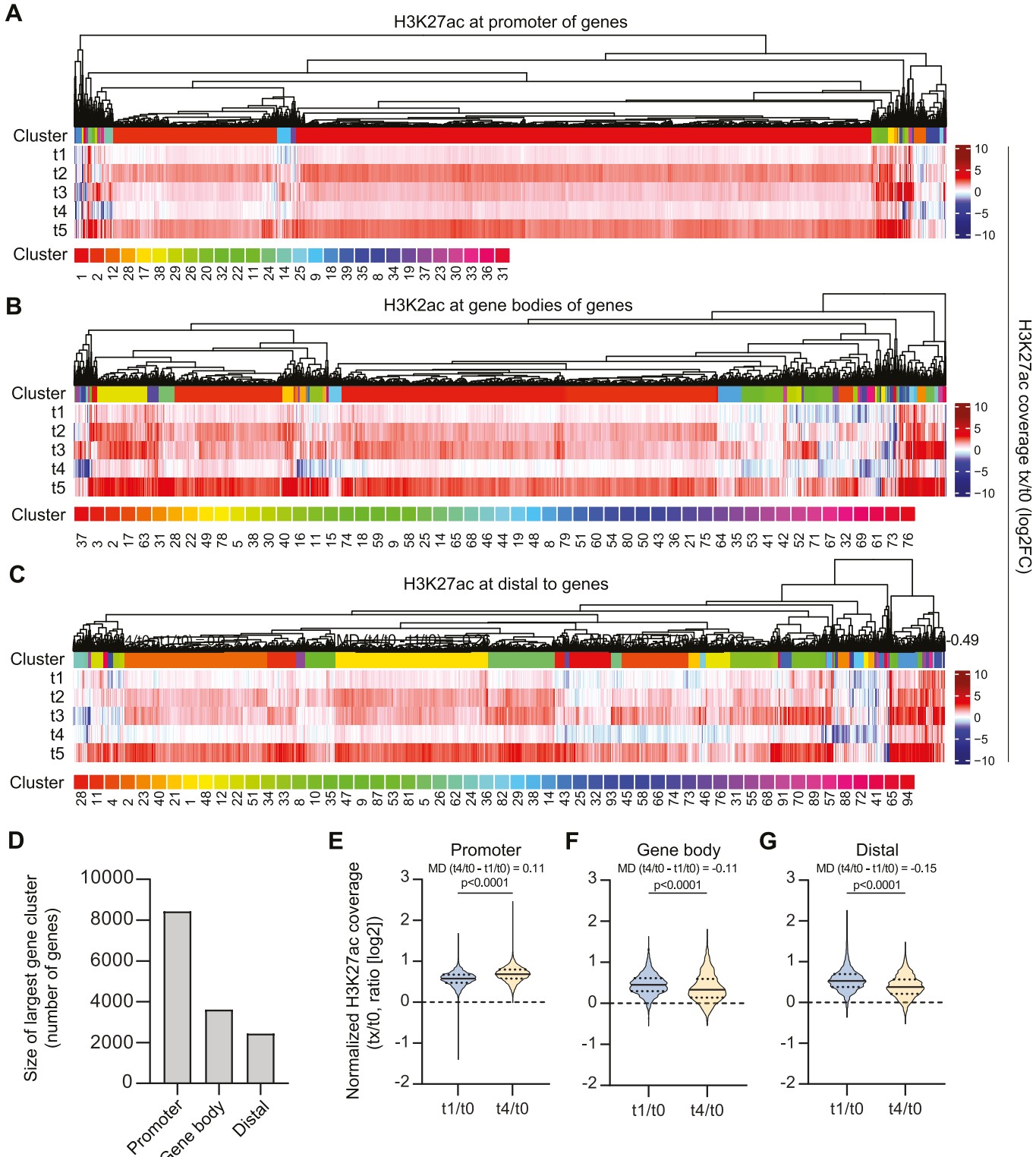

**Figure 4.  Gene clusters with similar H3K27ac alterations can be identified at promoters across different vaccination time points.**

Unsupervised identification of gene clusters exhibiting similar H3K27ac alterations across vaccinations at promotors (**A**) gene body (**B**) and distal to genes (**C**). (**D**) Size of largest gene clusters found in unsupervised hierarchical clustering of genes related to H3K27ac. (**E–G**) Comparative t1 vs t4 H3K27ac coverage ratios (log2) relative to t0 in promoters ($n = 8417$) (**E**), gene bodies ($n = 3606$) (**F**) and distal regions ($n = 2433$) (**G**). The median difference (MD) between the t4/t0 and t1/t0 ratios is shown for each genomic region. P values were calculated by using a Wilcoxon test. Values were highly significant ($P < 0.0001$) for all comparisons. P values less than 0.05 were considered statistically significant. Source data are available online for this figure.

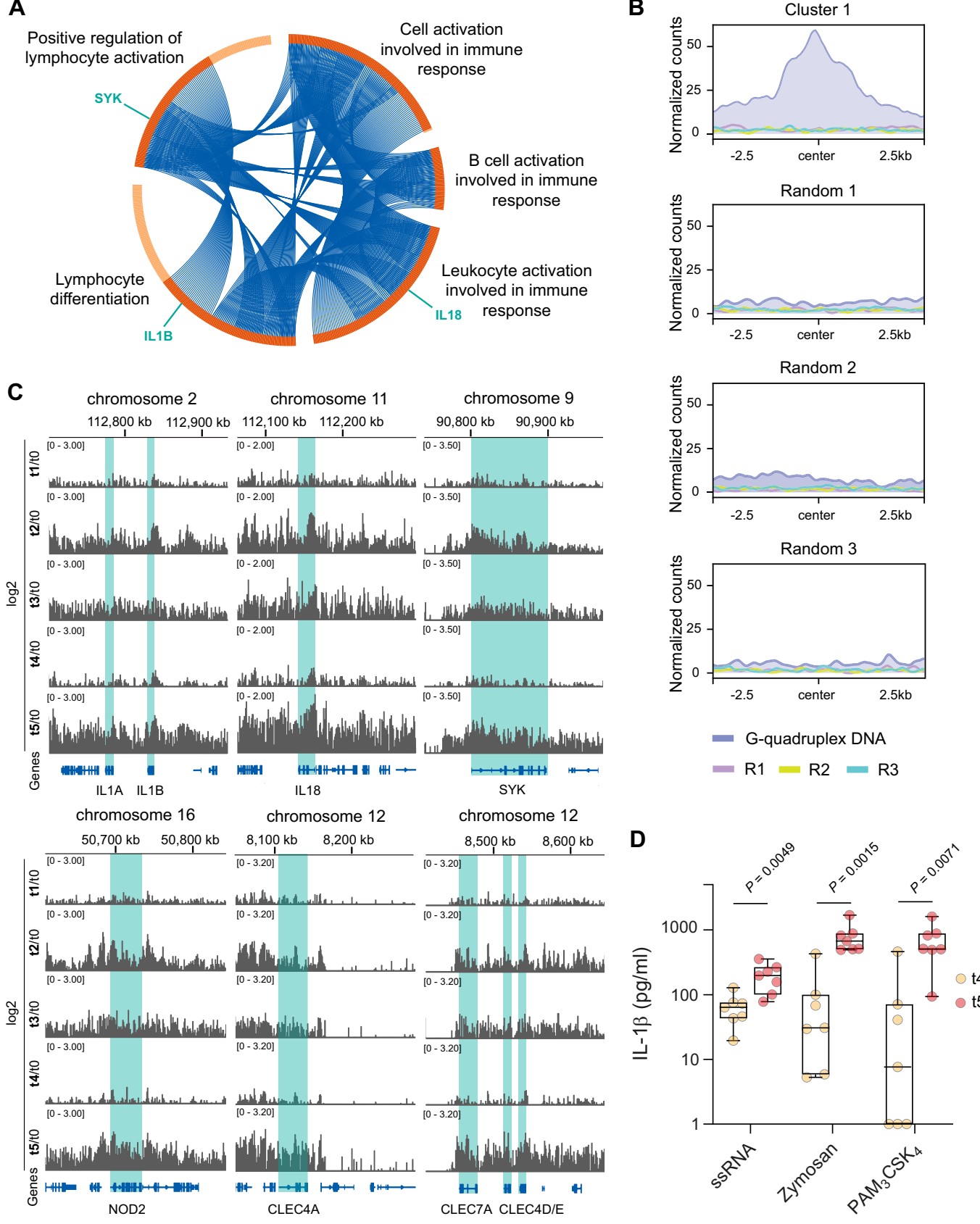

**Figure 5. H3K27ac in promoters of immune associated genes of Cluster 1.**

(A) Circos plot of selected immune-related terms identified in Cluster 1 (H3K27ac in promoters), illustrating the overlap of distinct genes across various GO terms. Dark red semicircles represent genes shared among multiple GO terms, whereas genes uniquely associated with specific GO terms are highlighted in light red. Individual genes of interest, including IL1B, IL-18, and SYK, are specifically marked as indicated. (B) G-quadruplex DNA from accessible chromatin of macrophages and its coverage in H3K27ac peaks located in promoters of immune genes part of cluster 1, and in matched randomized promoter sets. (C) Genome browser views of various chromosomal segments, displaying ratio (log2) of tx over t0 mean H3K27ac coverage across the time points. (D) Monocytes at t4 (n = 7; yellow) and t5 (n = 7; red) were isolated by CD14+ positive selection from PBMCs. Cells were seeded and incubated in the presence of M-CSF for 5 days. Differentiated cells were stimulated with ssRNA, Zymosan or Pam3Csk4 for 4 h. IL-1β secretion was quantified by ELISA. For statistical analysis, a multiple unpaired t test was used. Box plots indicate the median, the upper and lower quartile and the minimum and maximum values. Shown data points represent the technical mean of an independent experiment. P values less than 0.05 were considered statistically significant. Source data are available online for this figure.

and to attract transcription factors to collectively keep these promoters active (Esain-Garcia et al, 2024; Esnault et al, 2023; Hansel-Hertsch et al, 2020; Spiegel et al, 2021).

Our extensive study performed at five time points post vaccination revealed that it is primarily altered H3K27ac levels associated with genes coding for cytokines and innate immune receptors like C-type lectins which are shaped in a highly dynamic manner. In addition, we found H3K27ac on genes coding for major regulatory proteins of the human immune response. Overall, the data are in line with our recent ex vivo findings on SP-triggered signaling cascades which mediate inflammasome activation and cytokine secretion through activation of C-type lectins and SYK in macrophages of vaccinated individuals (Theobald et al, 2022).

Interestingly, epidemiological data indicate that live vaccines such as smallpox, measles and polio vaccine induce non-specific protective effects against infections other than the target diseases (Netea et al, 2020). Experiments performed in mice revealed that BCG vaccination induces epigenetic modifications, which alter both the receptor repertoire and functional state of circulating mononuclear cells leading to protection against fungal infection (Kleinnijenhuis et al, 2012). Similar effects were postulated for the BCG-vaccine ameliorating the clinical course of severe COVID-19 through enhancement of trained innate immunity in humans (Mantovani and Netea, 2020). However, a series of prospective clinical trials evaluating BCG for reduction of morbidity and mortality due to COVID-19 did not show a beneficial effect (Moorlag et al, 2022; Ten Doesschate et al, 2022; Upton et al, 2022). Nevertheless, it is intriguing to speculate that the broad and non-discriminative innate immune response we observed in macrophages following SARS-CoV-2 mRNA vaccination may induce resilience programs against other, non-related infectious diseases. More fundamental and clinical research is required to fully understand the translational potential of these findings.

To this end, our data on the immune response elicited by the third-dose booster vaccination are highly interesting. Initially, we were able to demonstrate that, in an immunologically naïve population, two mRNA vaccinations are required to mount epigenetic alterations and a potent innate immune response ex vivo. However, after a time span of 34 weeks (~6 months), administering a single booster vaccination dose was sufficient to induce significant epigenetic reprogramming of macrophages, resulting in the appearance of prominent H3K27ac marks on a large number of genes coding for a diverse repertoire of innate immune-associated genes. This could be correlated with the robust secretion of cytokines from macrophages stimulated with various PAMPs. The nature of this immune response was again non-specific and SARS-CoV-2 unrelated PAMPs were able to trigger macrophages post vaccination. Similar findings were recently made for SARS-CoV-2 infections which trigger long-lasting epigenetic and transcriptional modifications in macrophages, monocytes and

hematopoietic stem and progenitor cells (HSPC) (Cheong et al, 2023; Theobald et al, 2021). Thus, it is conceivable that macrophages of vaccinated or infected individuals may respond in a similar way toward viral infections other than the target disease. A limitation can be seen in the fact that, for our epigenetic studies, the mRNA-1273 vaccine construct was applied as the third-booster, whereas the first two vaccines were BNT162b2 mRNA vaccines. However, data evaluating the immune response of the two vaccine constructs are comparable, indicating that the choice of the vaccine has a minor impact on our epigenetic and transcriptomics findings (Naranbhai et al, 2022).

Overall, our data contradict findings made in another study, which postulates that immune memory following two consecutive mRNA vaccinations is short-lived with rapid development of immune homeostasis after a few weeks (Yamaguchi et al, 2022). While it is true that epigenetic marks decline over time, our study reveals that monocyte-derived macrophages remain altered and highly responsive upon restimulation. Promoter-associated H3K27ac marks were persistent several months after the first two vaccinations clearly indicating long-lived innate immune memory. We assume that it is these persistent marks that drive profound epigenetic alterations and strong immune responsiveness in macrophages following a single, third-dose booster vaccination.

The dynamic aspects we observed for epigenetic reprogramming of classical innate immune cells may have implications for the improvement of vaccine responses and designs, the overall immune response towards non-related infections and even our understanding of post vaccination inflammatory diseases which occur in a small number of vaccinated individuals. Thus, the results of this study advance our current understanding of mRNA-based vaccination and imply important considerations for the development of mRNA-based vaccines in the future.

## Methods

**Reagents and tools table**

| Reagent/resource | Reference or source | Identifier or catalog number |
|---|---|---|
| **Experimental models** | | |
| Patient-derived macrophages | This study | N/A |
| **Recombinant DNA** | | |
| S ectodomain coding pαH plasmid | Wrapp et al, 2020; Hsieh et al, 2020 | N/A |
| **Antibodies** | | |
| H3K27ac | Active Motif | Cat# 39133 |

| Reagent/resource | Reference or source | Identifier or catalog number |
|---|---|---|
| **Chemicals, enzymes, and other reagents** | | |
| Bio-Mag Plus Concanavalin A coated beads | Polysciences | Cat# 86057 |
| CD14 MicroBeads, human - lyophilized | Miltenyi Biotec | Cat# 130-097-052 |
| CUTANA pAG-MNase | EpiCypher | Cat# 15-1016 |
| Ficoll Paque Plus | GE Healthcare | Cat# 17-1440-02 |
| KINK-1 | Sigma-Aldrich | Cat# SML2098-5MG |
| MCC950 | Sigma-Aldrich | Cat# 5381200001 |
| M-CSF, research grade (human) | Miltenyi Biotec | Cat# 130-096-491 |
| Nigericin | Sigma-Aldrich | Cat# N7143 |
| ORN06/LyoVec™ | InvivoGen | Cat# tlrl-orn6 |
| Pam3CSK4 | InvivoGen | Cat# tlrl-pms |
| SARS-CoV-2 spike protein | This study | N/A |
| ssRNA | InvivoGen | |
| Zymosan | InvivoGen | Cat# tlrl-zyn |
| **Software** | | |
| Adobe Illustrator v26.4.1 | Adobe | N/A |
| bcl2fastq2 v2.20.0 | Illumina | N/A |
| bedtools v2.31 | Quinlan and Hall, 2010 | N/A |
| Bowtie2 v2.4.5 | Langmead and Salzberg, 2012 | N/A |
| ChIPpeakAnno v3.34 | Love et al, 2014 | N/A |
| clusterProfiler v4.0.5 | Yu et al, 2012 | N/A |
| DESeq2 v1.32.0 | Bioconductor | N/A |
| enrichplot v1.12.2 | Bioconductor | N/A |
| EnhancedVolcano v1.13.2 | Bioconductor | N/A |
| Excel | Microsoft | N/A |
| fgsea v1.18.0 | Bioconductor | N/A |
| GraphPad Prism v9.5.1 | GraphPad | N/A |
| LinRegPCR | Amsterdam UMC | N/A |
| Metascape | Zhou et al, 2019 | N/A |
| nf-core RNA-seq pipeline v3.0 | Ewels et al, 2020 | N/A |
| plotProfile v3.5.4 | deepTools | N/A |
| R version v4.1.1 (2021-08-10) | R Core Team 2021 | N/A |
| Samtools v1.17 | Li et al, 2009 | N/A |
| SEACR v1.3 | Meers et al, 2019 | N/A |
| xPONENT | Diasorin | N/A |
| **Other** | | |
| Human IL-1 beta Uncoated ELISA Kit | Thermo Fisher Scientific | Cat# 88-7261-88 |
| Human Luminex® Discovery Assays | Biotechne | Cat# LXSAHM |
| KAPA Library Quantification Kit | KAPA Biosystems | Cat# KK4824 |

| Reagent/resource | Reference or source | Identifier or catalog number |
|---|---|---|
| mirVana miRNA isolation kit | Thermo Fisher Scientific | Cat# AM1561 |
| TruSeq DNA nano Kit | Illumina | Cat# 20015964 |

## Isolation of monocytes and differentiation to macrophages

Blood samples were obtained at different time points from healthy, unvaccinated donors or donors that had received one, two or three doses of SARS-CoV-2 mRNA vaccine (Comirnaty [Pfizer, New York City, NY, USA/BioNTech SE, Mainz, Germany] or/and, Spikevax [Moderna, Inc., Cambridge, MA, USA], respectively. For all human samples, written informed consent was obtained in accordance with the declaration of Helsinki and the experiments conformed to the principles set out in the Department of Health and Human Services Belmont Report. The study was approved by the ethics committee of Cologne (Reference number 21-1283). Only adults were included in the study. Individuals with COVID-19 infections before or after vaccination were excluded.

PBMCs (peripheral blood mononuclear cells) were purified by density gradient centrifugation (Ficoll Plus, GE Healthcare, Chicago, IL, USA). CD14$^+$ cells were isolated from PBMCs by positive selection (Miltenyi Biotech, Bergisch Gladbach, Germany). $2.5 \times 10^5/5 \times 10^4$ CD14$^+$ cells were seeded into 24/96-well plates (TPP Techno Plastic Products AG, Trasadingen, Switzerland) and cultured for 5 days in Roswell Park Memorial Institute (RPMI) 1640 Medium (Thermo Fisher Scientific, Waltham, MA, USA) containing 10% fetal bovine serum (Thermo Fisher Scientific) and 50 ng/ml M-CSF (Miltenyi Biotec) for macrophage differentiation at 37 °C and 5% CO$_2$.

## Ex vivo stimulation of macrophages

Prior experiments medium of differentiated macrophages was exchanged, and macrophages were incubated for a further 2 h at 37 °C and 5% CO$_2$. After medium exchange, SARS-CoV-2 protein (0.1 µg/ml), which was recombinantly expressed as previously described (Theobald et al, 2021), zymosan (10 µg/ml; Invivogen, Toulouse, France), or ssRNA (8 µg/ml, ORN06/LyoVec™, Invivogen) were added for 4 h. Subsequently, nigericin (5 µM) (Sigma-Aldrich) was added for 2 h at 37 °C and 5% CO2. For experiments with inhibitors, cells were incubated prior first stimulation with either DMSO (Sigma-Aldrich), MCC950 (10 µM) (Sigma-Aldrich) or KINK-1 (10 µM) (Sigma-Aldrich). All assays were performed in technical duplicates. Supernatants were frozen at −80 °C for subsequent cytokine analysis.

## Quantification of cytokines

Quantitative detection of IL-1β was achieved using the IL-1 beta Human Uncoated ELISA Kit (Thermo Fisher Scientific) and performed according to the manufacturer's instructions. Supernatants of primary macrophages were diluted in ELISA diluent 1:5–1:50. All samples were measured in technical duplicates and absorbance was measured in a microplate reader (Hidex Oy, Turku, Finland). For other cytokines, a Cytokine array was performed via Luminex Discovery Assay (R&D Systems, Minneapolis, MN, USA) with the indicated analytes according to the manufacturer's instructions. Samples were centrifuged at $1000 \times g$

for 10 min and diluted 1:4 in Calibrator Diluent (R&D Systems) prior to analysis. Cytokines/Chemokines were measured with Luminex 200 xMAP system (Luminex) and quantified by comparison to a standard curve. xPONENT software was used for data collection and analysis.

## Transcriptome sequencing (RNA-Seq) analyses

Human macrophages were isolated as described before and seeded into 24-well plates with $2.5 \times 10^5$ cells per well. Stimulation was performed as described before. In brief, cells were washed 2× with DPBS and isolation of the RNA was performed with the mirVana miRNA isolation kit (Thermo Fisher Scientific) according to the manufacturer's instructions. RNA-Seq library prep was performed with 100 ng total RNA input and the NEBNext Ultra RNA library prep protocol (New England Biolabs, Ipswich, MA, USA) according to standard procedures. Libraries were validated and quantified (Tape Station 4200, Agilent Technologies, Santa Clara, CA, USA).

All libraries were quantified by using the KAPA Library Quantification Kit (Roche, Basel, Switzerland) and the 7900HT Sequence Detection System (Applied Biosystems, Foster City, CA, USA). Sequencing was done with NovaSeq6000 sequencers (Illumina, San Diego, CA, USA) with a PE100bp read length aiming at 50 M reads/sample (RNA-Seq) or a SR50bp read length aiming at 5 M reads/sample (small RNA). Demultiplexing and FastQ file generation were performed using Illumina's bcl2fastq2 software (v2.20.0).

RNA-seq was performed with a directional protocol. Quality control, trimming, and alignment were performed using the nf-core RNA-seq pipeline (v3.0) (Ewels et al, 2020). Details of the software and dependencies for this pipeline can be found at https://github.com/nf-core/rnaseq/blob/master/CITATIONS.md. The reference genome sequence and transcript annotation used were Homo sapiens genome GRCh38 from Ensembl version 103. Differential expression analysis was performed in R version 4.1.1 (2021-08-10) (R Core Team 2021) with DESeq2 v1.32.0 to make pairwise comparisons between groups. Log Fold Change shrinkage estimation was performed with ashr (Stephens, 2017). Only genes with a minimum coverage of 10 reads in 6 or more samples from each pairwise comparison were considered as candidates to be differentially expressed. Genes were considered differentially expressed if they showed a log2 (Fold Change) > 1 and were below an FDR of 0.05. Genes with a minimum coverage of 10 reads in 6 or more samples from each pairwise comparison were included in functional enrichment analyses and considered as the 'gene universe' for over-representation-based analyses. Functional enrichment analysis was performed with clusterProfiler v4.0.5 (Yu et al, 2012). Gene Set Enrichment Analysis (GSEA) was performed using the fgsea v1.18.0 algorithm (Subramanian et al, 2005). The p-value cutoff was 0.05 by permutation, (using the default values in clusterProfiler), for genes rank ordered by log2 (fold change).

The GSEA dotplot was plotted with enrichplot v1.12.2. The Volcano plot was plotted with a modified version of the EnhancedVolcano function from EnhancedVolcano v1.13.2.

## CUT&RUN sample preparation

CUT&RUN was performed on 150 K macrophages per sample adapting a previously described protocol (Skene and Henikoff, 2017). In brief, cells were washed twice with Wash Buffer (20 mM MHEPES, 150 mM NaCl, 0.5 mM Spermidine) and bound to activated Concanavalin A beads (Polysciences Inc., Warrington, PA, USA) for 10 min at room temperature. Cell-bead suspension was then resuspended in antibody buffer (20 mM MHEPES, 150 mM NaCl, 0.5 mM Spermidine, 0.05% Digitonin, 2 mM EDTA) and incubated overnight at 4 °C with a H3K27ac antibody (Active Motif, Carlsbad, CA, USA #39133) (dilution 1:50). Cell-bead suspension was then washed twice with Digitonin Buffer (20 mM MHEPES, 150 mM NaCl, 0.5 mM Spermidine, 0.05% Digitonin) and incubated with 2.5 µl CUTANA pAG-MNase (EpiCypher, Durham, NC, USA), for 10 min. After washing samples twice with Digitonin buffer, 1 µl 100 mM CaCl2 was added to samples which were incubated for 2 h at 4 °C, rotating. To stop the reaction, STOP buffer (340 mM Nacl, 20 mm EDTA, 4 mM EGTA, 0.02% Digitonin, 50 µg/ml RNase A, 50 µg/ml Glycogen, 50 pg sheared genomic DNA of *Saccharomyces cerevisiae* as spike-in control) was added in each tube. Samples were incubated at 37 °C for 10 min at 500 rpm, and after centrifugation, liquid was collected, DNA was purified using aDNA clean concentrator kit (Zymo Research, Irvine, CA, USA), and DNA was eluted in elution buffer. For library preparation, we used the TruSeq DNA nano kit and protocol from Illumina with 15 cycles of PCR. Libraries were validated and quantified (Agilent Tape Station 4200). A pool of all libraries was quantified by using the KAPA Library Quantification Kit and the Applied Biosystems 7900HT Sequence Detection System. Sequencing was done with NovaSeq sequencers (Illumina) and a PE100bp read length aiming at 10 M reads/sample.

## CUT&RUN data analysis

All data analysis was performed using the European Galaxy server (https://usegalaxy.eu). Raw fastq files were mapped and treated as described by Zheng et al (https://www.protocols.io/view/cut-amp-tag-data-processing-and-analysis-tutorial-e6nvw93x7gmk/v1). Paired-end reads were aligned using Bowtie2 (version 2.4.5), Samtools (version 1.17) and bedtools (version 2.31) using the following parameters: --end-to-end --very-sensitive --no-mixed --no-discordant --phred33 -I 10 -X 700 for mapping of inserts 10-700 bp in length (Langmead and Salzberg, 2012; Li et al, 2009; Quinlan and Hall, 2010). We used the GRCh38 human reference genome for alignment and the genome sacCer3 of *S. cerevisiae* for spike-in calibration after adjusting all human libraries to the same read depth For peak calling SEACR (version 1.3) was used; fragment counts were normalized using the spike-in read count (Meers et al, 2019). Thus, the normalization option of SEACR was set to "non"; the numeric threshold used was 0.01.

### *Genome-wide coverage ratios of H3K27ac at t1 to t5 relative to t0 in promoters, gene bodies, and distal regions*

We quantified normalized genome-wide coverage for all 41 H3K27ac libraries using coverageBED (version 2.31.1) and calculated mean coverage for each time point across the libraries. Promoters were defined as regions spanning -1 kb upstream of the transcription start site (TSS) to 250 bp downstream of the TSS. Gene bodies were defined as 251 bp downstream of the TSS to the transcription end site (TES). Distal regions were defined as areas located more than 1 kb and up to 100 kb away from genes (from TSS to TES). H3K27ac peaks were called for all 41 samples, concatenated, sorted using sortBED (version 2.31.1), merged with mergeBED (version 2.31.1), and filtered to retain peaks with canonical chromosomal annotations (grep pattern: chr([0-9XYxy]+)\b). To calculate genome-wide ratios of mean H3K27ac coverage at time points T1 to T5 relative to T0, we computed the ratio of the sum of mean coverage at each time point to T0 for each H3K27ac peak (bedtools_unionbedgraph, version 2.31.1;

tp_awk_tool, version 9.3). The resulting H3K27ac peaks and their associated ratios were further annotated to promoters, gene bodies, and distal regions using closestBed (version 2.31.1).

### Unsupervised identification of gene clusters exhibiting similar H3K27ac alterations across time points

To derive consistent values for each gene, we averaged the H3K27ac coverage ratios across peaks associated with each gene feature (promoter, gene body, or distal region) for each time point relative to t0. To resolve the complexity of coverage ratios across time points, we employed dynamic hierarchical clustering to identify gene clusters with distinct density, size, and structure (dynamicTreeCut, version 1.63-1). Clusters with a positive average silhouette score (cluster, version 2.1.6; code is provided in the Data availability section) were retained for re-clustering, visual representation (Fig. 4A), and downstream analyses such as gene ontology (clusterProfiler, version 4.12.6).

### G-quadruplex DNA sequence profiling in H3K27ac peaks that overlap with accessible chromatin of macrophages

We considered experimentally validated human G4 DNA sequences overlapping annotated accessible chromatin in macrophages. Annotated locations of accessible chromatin (from DNase-seq and ATAC-seq datasets) were retrieved from CHIP-ATLAS using the peak annotation browser (Zou et al, 2022). To profile G4 DNA coverage in H3K27ac peaks, G4 DNA sequence annotations were converted into binary coverage files (bigwig format) and analyzed with deeptools (plotProfile, version 3.5.4; code is provided in the Data availability section). G4 DNA coverage was profiled in H3K27ac peaks annotated to promoters, gene bodies, and distal regions, as well as in promoters of immune genes from Cluster 1, which were associated with immune-related terms.

### H3K27ac coverage profiling in promoters of immune genes

We prepared two sets of H3K27ac peaks overlapping promoters of genes that: (a) exhibited increased expression in response to SP stimulation ($log2FC > 1$, $FDR < 0.05$), and (b) were immune-associated genes from Cluster 1, annotated with immune-related GO terms such as immune, inflammatory, cytokine, chemokine, leukocyte, and lymphocyte. For further details, please refer to the code provided in "Data availability". Coverage profiles for these peaks were generated using deeptools (plotProfile, version 3.5.4).

For gene ontology, circus plot and Kegg pathway enrichment, ChIPpeakAnno (version 3.34) or Metascape were used (Love et al, 2014; Zhou et al, 2019).

### Statistical analysis

Statistical analysis was performed with GraphPad Prism 9.5.1 software (GraphPad). Statistical parameters (value of n, statistical calculation, etc.) are also provided in the figure legend. $P$ values less than or equal to 0.05 were considered statistically significant. Statistical tests were used as indicated in the figure legends. Box plots indicate the median and the upper and lower quartile. Outliers are plotted as individual dots (outside the 10–90 percentile). Scatter dot plots show mean. Data points representing biological replicates. All experiments were conducted at least in technical duplicates. Samples between the different groups were matched regarding age, sex and vaccination status. Apart from that all samples were selected randomly, and all other information were blinded. Sample sizes were chosen depending on the expected variance and available samples.

### Graphics

Graphics were created with BioRender.com and Adobe Illustrator.

## Data availability

The data generated during this study are deposited at the European Genome-phenome Archive (EGA) under access number EGAS50000000341, which is hosted by the EBI and the CRG (https://ega-archive.org/studies/EGAS50000000341). Computer code is available at https://github.com/HaenselHertschEpiLab/Epigenetic_memory_of_SARS-CoV-2_vaccination_in_macrophages.

The source data of this paper are collected in the following database record: biostudies:S-SCDT-10_1038-S44320-025-00093-6.

## Peer review information

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

## Acknowledgements

AS is supported by the German Federal Ministry of Education and Research (BMBF) (01KI2108) (Junior Research Groups Infectious Diseases), a fellowship of the Cologne Clinician Scientist Program (CCSP), funded by the German Research Council (DFG) (FI 773/15-1), and the Career Advancement Groups Program of the Center of Molecular Medicine Cologne, Faculty of Medicine and University Hospital of Cologne, University of Cologne. JR is funded by the DFG (SFB1403), the German Center for Infection Research (DZIF; TTU-TB grants 02.913 and 02.814), BMBF (IdEpiCo), the European Union Horizon 2020 program (ERA4TB) and COVIM, part of the "Network of University Medicine (NUM)", funded by the Federal Ministry of Education and Research (BMBF) ("NUM 2.0" grant no: 01KX2121). ST and JR are supported by a research grant of the Center for Molecular Medicine Cologne (CMMC - B10), and ST by a stipend from the Imhoff-Stiftung and Cologne Fortune. Work in the PT lab is funded by the Max Planck Society and funds of the German Excellence Strategy (CECAD; EXC 2030-390661388). RH-H receives funding through Center of Molecular Medicine Cologne, DFG CRC1399 (INST 216/1057-2), Fritz Thyssen Foundation (10.22.1.010MN), CANTAR which is funded through the program "Netzwerke 2021", an initiative of the Ministry of Culture and Science of the State of Northrhine Westphalia, DFG HA 8562/4-1 and DFG RU5504 (HA 8562/5-1) and DFG CRC1678 (INST 216/1317-1). The authors would like to thank Dr. Tony Müller (www.tru-id.de) for critical reading and the preparation of the manuscript.

## Author contributions

**Alexander Simonis**: Conceptualization; Data curation; Formal analysis; Supervision; Funding acquisition; Validation; Investigation; Visualization; Methodology; Writing—original draft; Project administration. **Sebastian J Theobald**: Conceptualization; Data curation; Formal analysis; Supervision; Validation; Investigation; Visualization; Methodology; Writing—original draft; Project administration. **Anna Eva Koch**: Formal analysis; Investigation; Visualization; Methodology. **Ram Mummadavarapu**: Data curation; Formal analysis; Validation; Investigation; Methodology; Writing—review and editing. **Julie M Mudler**: Formal analysis; Investigation; Writing—review and editing. **Andromachi Pouikli**: Investigation; Methodology; Writing—review and editing. **Ulrike Göbel**: Resources; Data curation; Software; Formal analysis; Validation; Investigation; Visualization; Methodology; Writing—review and editing. **Richard Acton**: Software; Formal analysis; Validation; Investigation; Visualization; Methodology; Writing—review and editing. **Sandra Winter**: Investigation. **Alexandra Albus**: Formal analysis; Investigation; Writing—review and editing. **Dmitriy Holzmann**: Investigation. **Marie-Christine Albert**: Investigation. **Michael Hallek**: Resources; Writing—review and editing. **Henning Walczak**: Resources; Writing—review and editing. **Thomas Ulas**: Resources; Data curation; Software. **Manuel Koch**: Investigation; Writing—review and editing. **Peter Tessarz**: Conceptualization; Resources; Data curation; Formal analysis; Supervision; Funding acquisition; Validation; Investigation; Methodology; Writing—original draft; Project administration. **Robert Hänsel-Hertsch**: Conceptualization; Resources; Software; Formal analysis; Supervision; Funding acquisition; Validation; Investigation; Visualization; Methodology; Writing—original draft; Project administration. **Jan Rybniker**: Conceptualization; Resources; Supervision; Funding acquisition; Validation; Methodology; Writing—original draft; Project administration.

Source data underlying figure panels in this paper may have individual authorship assigned. Where available, figure panel/source data authorship is listed in the following database record: biostudies:S-SCDT-10_1038-S44320-025-00093-6.

## Funding

## Disclosure and competing interests statement

The authors declare no competing interests.

# Expanded View Figures

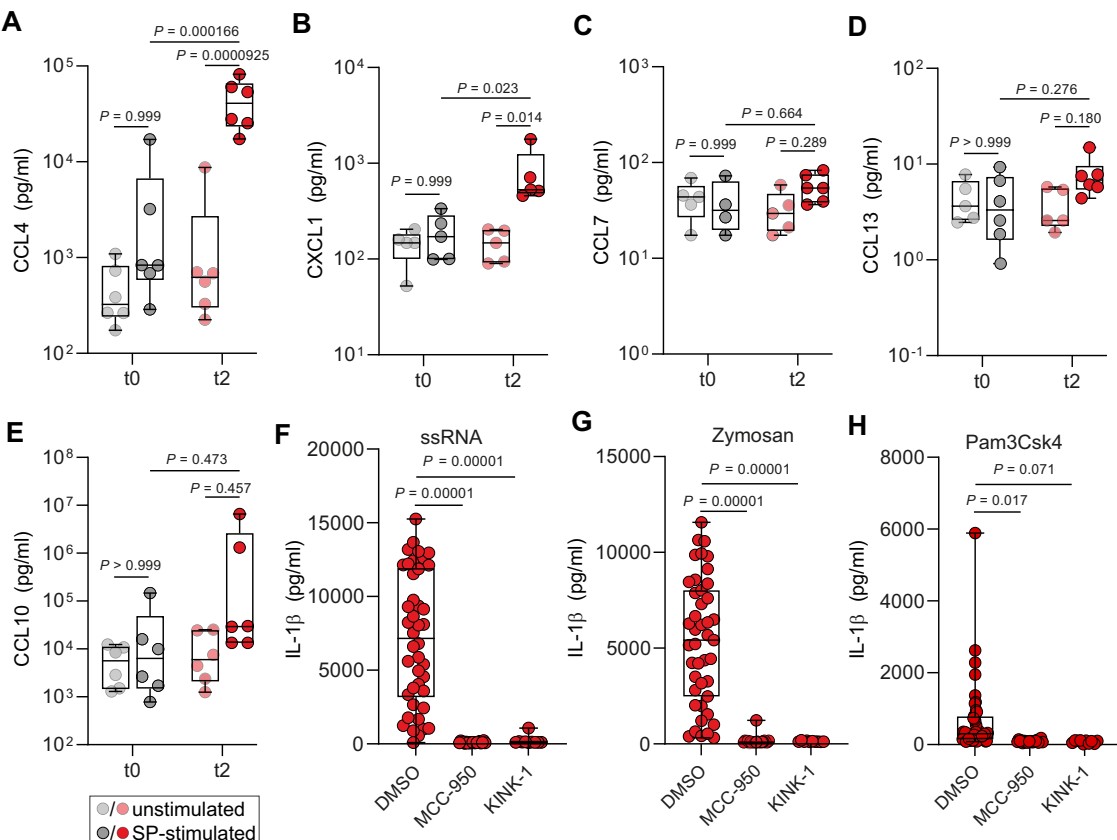

**Figure EV1. Inflammasome dependent secretion of IL-1β in macrophages following vaccination.**

(**A**) Monocytes were isolated by CD14+ selection from blood samples from unvaccinated (t0) ($n = 6$) (gray) and vaccinated individuals (t2) ($n = 6$) (red) and seeded and incubated in the presence of M-CSF for 5 days. Differentiated macrophages were stimulated with SP (t0: dark gray dots; t2: dark red dots) or left unstimulated (t0: light gray dots; t2: light red dots). Concentrations of CCL4, CXCL1 (**B**), CCL7 (**C**), CXCL13 (**D**) and CXCL10 (**E**), were measured by multiplex analyses. For statistical analysis, two-way ANOVA with Sidak's multiple comparison analysis was used. Box plots indicate the median, the upper and lower quartile and the minimum and maximum values. (**F**) Monocyte-derived macrophages from donors before (t0) and 2 weeks after second vaccination (t2) were generated as described in (**A**). Differentiated macrophages were stimulated with ssRNA, Zymosan (**G**) or Pam₃Csk₄ (**H**) for 4 h in the presence of MCC950 (10 µM) ($n = 20$), KINK-1 (5 µM) ($n = 10$) or left untreated. IL-1β secretion was quantified by ELISA. For statistical analysis, one-way ANOVA with Dunnett's multiple comparison test comparing MCC950/KINK-1-treated cells to untreated cells was used. Box plots indicate the median, the upper and lower quartile and the minimum and maximum values. Shown data points represent the technical mean of an independent experiment. *P* values less than 0.05 were considered statistically significant.

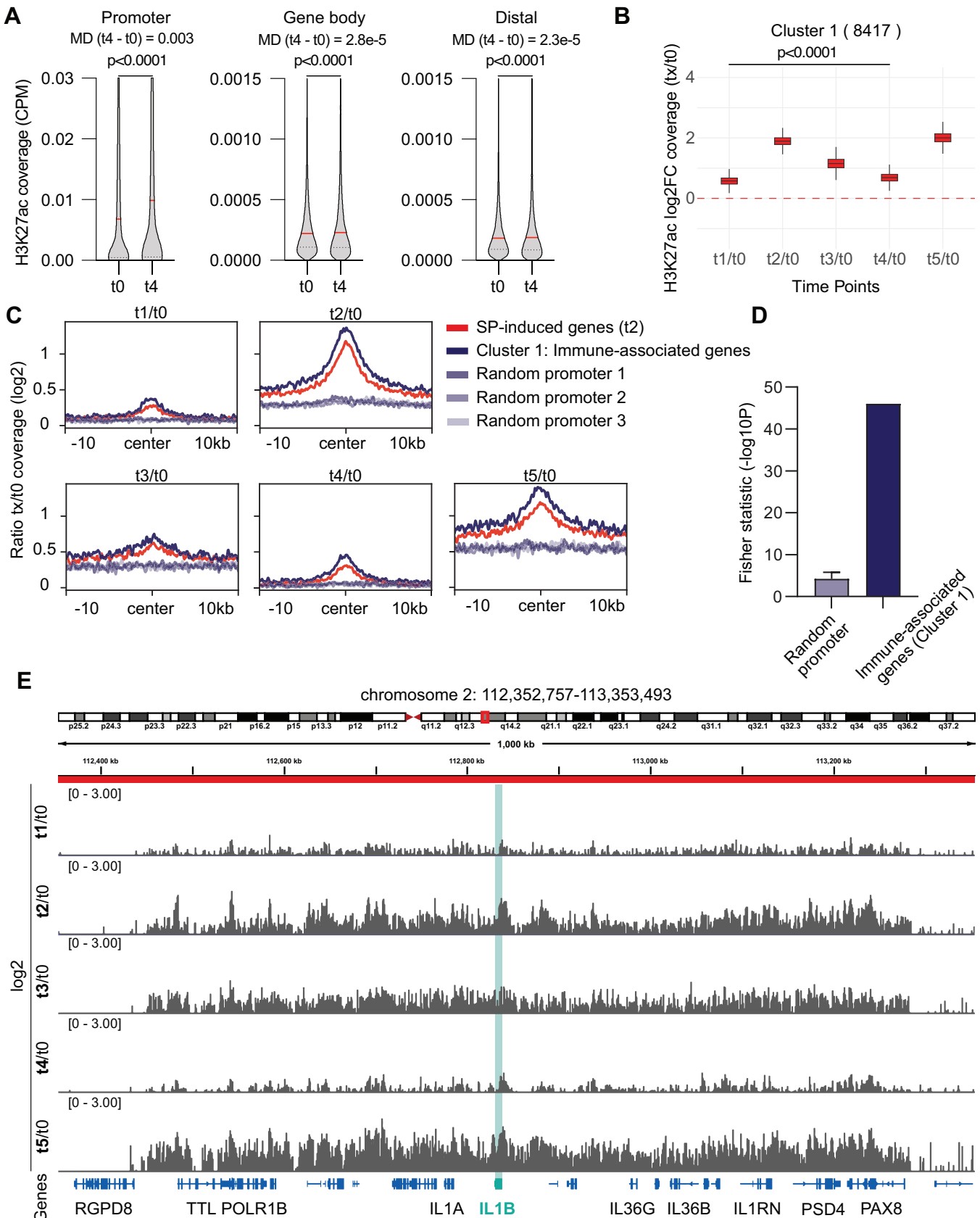

**Figure EV2. H3K27ac coverage across various gene regions and clusters.**

(A) Distribution of normalized (counts per million) H3K27ac coverage in peaks of promoters (1 kb upstream to 250 bp downstream of transcription start sites [TSS]) (left), gene bodies (251 bp downstream of TSS to transcription end sites [TES]) (middle) and distal regions (1 kb to 100 kb away from TSS and TES) (right). All distribution comparisons are significantly different ($P < 0.0001$). $P$ values were calculated using a Wilcoxon test. $P$ values less than 0.05 were considered statistically significant. Median difference (MD) between t4 and t0 is indicated for all three genomic regions. (B) H3K27ac coverage shown as fold change (log2) across time points (t1-t5) relative to t0 for all genes part of Cluster 1. (C) Fold change of H3K27ac read coverage in tx relative to t0 in H3K27ac peaks overlapping with selected gene promoter from SP-stimulated genes ($n = 576$), Cluster 1 immune-associated genes ($n = 333$), Random promoter 1 ($n = 333$), Random promoter 2 ($n = 333$), Random promoter 3 ($n = 333$). (D) Fisher statistic ($-\log 10P$) of H3K27ac peaks in SP-induced gene promoter overlapping with H3K27ac peaks in Cluster 1 immune gene promoter or H3K27ac peaks in random promoter. Error bar indicates standard deviation of the fisher statistic from the 3 randomly distributed H3K27ac peaks in all annotated promoters. (E) Genome browser view of 1000 kb of chromosome 2, displaying ratio (log2) of tx over t0 mean H3K27ac coverage across the time points.

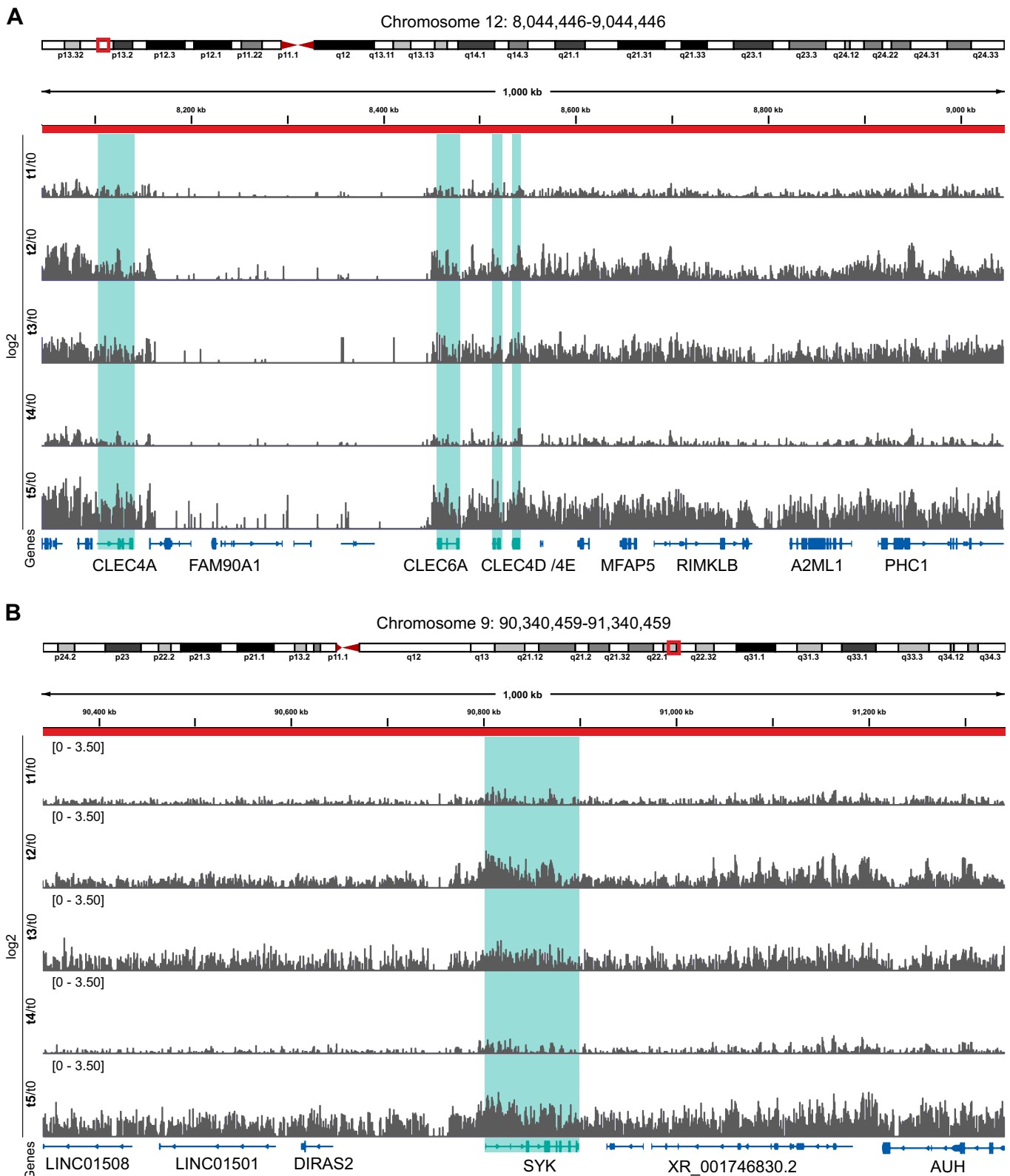

**Figure EV3. H3K27ac coverage across different vaccination time points.**

(A) Genome browser view of 1000 kb of chromosome 12 and 9 (B), displaying ratio (log2) of tx over t0 mean H3K27ac coverage across the time points.

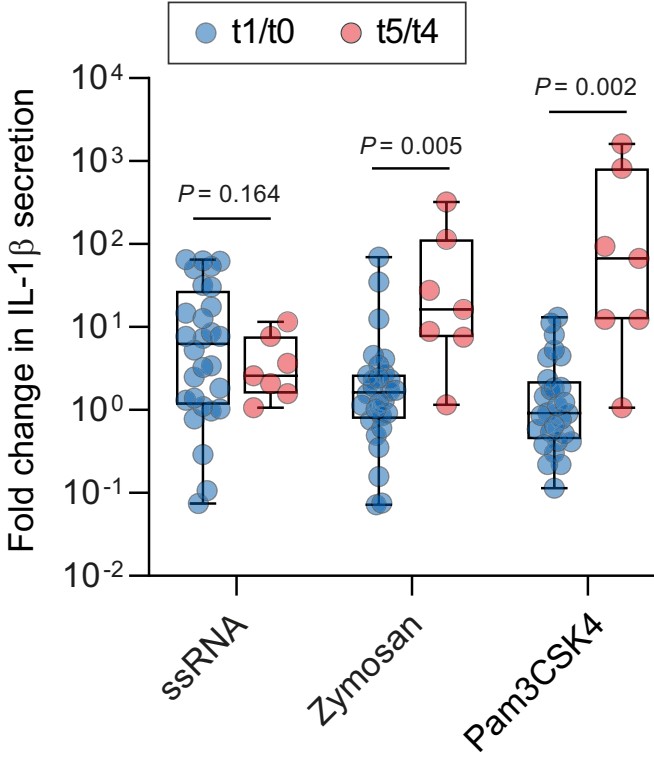

**Figure EV4.   Cytokine release before and after the third dose booster vaccination.**

Fold change of L-1β release upon stimulation with ssRNA, zymosan or Pam3CSK4 between t1/t0 ($n = 28$) (blue) and t5/t4 ($n = 7$) (red). For statistical analysis, a multiple unpaired *t* test was used. Box plots indicate the median, the upper and lower quartile and the minimum and maximum values. Shown data points represent the technical mean of an independent experiment.

