## [Peer Review File · Molecular Systems Biology]

Persistent epigenetic memory of SARS-CoV-2 mRNA vaccination in monocyte-derived macrophages

Alexander Simonis, Sebastian Theobald, Anna Koch, Ram Mummadavarapu, Julie Mudler, Andromachi Pouikli, Ulrike Göbel, Richard Acton, Sandra Winter, Alexandra Albus, Dmitriy Holzmann, Marie-Christine Albert, Michael Hallek, Henning Walczak, Thomas Ulas, Manuel Koch, Peter Tessarz, Robert Hänsel-Hertsch, and Jan Rybniker

Corresponding author(s): Jan Rybniker (jan.rybniker@uk-koeln.de) , Robert Hänsel-Hertsch (robert.haensel-hertsch@uni-koeln.de)

Review Timeline:

Submission Date:	15th Jun 23
Editorial Decision:	22nd Aug 23
Revision Received:	10th Dec 24
Editorial Decision:	21st Jan 25
Revision Received:	10th Feb 25
Accepted:	24th Feb 25

Editors: Poonam Bheda and Jingyi Hou

Transaction Report:

22nd Aug 2023

Dear Dr. Rybniker,

Thank you for the submission of your manuscript to Molecular Systems Biology. We have now received feedback from two reviewers who agreed to evaluate your manuscript. As you will see from the reports below, the referees acknowledge the interest of the study. However, addressing the reviewers' concerns in full will be necessary for further considering the manuscript in our journal, and acceptance of the manuscript will entail a second round of review. Molecular Systems Biology encourages a single round of revision only and therefore, acceptance or rejection of the manuscript will depend on the completeness of your responses included in the next, final version of the manuscript. For this reason, and to save you from any frustrations in the end, I would strongly advise against returning an incomplete revision.

We are expecting your revised manuscript within three months, if you anticipate any delay, please contact us.

We require:

4) A .docx formatted letter INCLUDING the reviewers' reports and your detailed point-by-point responses to their comments. As part of the EMBO Press transparent editorial process, the point-by-point response is part of the Review Process File (RPF), which will be published alongside your paper.

5) A complete author checklist, which you can download from our author guidelines (<https://www.embopress.org/page/journal/17574684/authorguide#submissionofrevisions>). Please insert information in the checklist that is also reflected in the manuscript. The completed author checklist will also be part of the RPF.

6) Please note that all corresponding authors are required to supply an ORCID ID for their name upon submission of a revised manuscript.

7) It is mandatory to include a 'Data Availability' section after the Materials and Methods. Before submitting your revision, primary datasets produced in this study need to be deposited in an appropriate public database, and the accession numbers and database listed under 'Data Availability'. Please remember to provide a reviewer password if the datasets are not yet public (see <https://www.embopress.org/page/journal/17574684/authorguide#dataavailability>).

In case you have no data that requires deposition in a public database, please state so in this section. Note that the Data Availability Section is restricted to new primary data that are part of this study. This study includes no data deposited in external repositories.

8) For data quantification: please specify the name of the statistical test used to generate error bars and P values, the number (n) of independent experiments (specify technical or biological replicates) underlying each data point and the test used to calculate p-values in each figure legend. The figure legends should contain a basic description of n, P and the test applied. Graphs must include a description of the bars and the error bars (s.d., s.e.m.). Please provide exact p values.

9) Our journal encourages inclusion of *data citations in the reference list* to directly cite datasets that were re-used and obtained from public databases. Data citations in the article text are distinct from normal bibliographical citations and should directly link to the database records from which the data can be accessed. In the main text, data citations are formatted as follows: "Data ref: Smith et al, 2001" or "Data ref: NCBI Sequence Read Archive PRJNA342805, 2017". In the Reference list, data citations must be labeled with "[DATASET]". A data reference must provide the database name, accession

number/identifiers and a resolvable link to the landing page from which the data can be accessed at the end of the reference. Further instructions are available at .

<https://www.embopress.org/page/journal/17574684/authorguide#expandedview>

11) For more information: There is space at the end of each article to list relevant web links for further consultation by our readers. Could you identify some relevant ones and provide such information as well? Some examples are patient associations, relevant databases, OMIM/proteins/genes links, author's websites, etc...

12) Author contributions: CRediT has replaced the traditional author contributions section because it offers a systematic machine readable author contributions format that allows for more effective research assessment. Please remove the Authors Contributions from the manuscript and use the free text boxes beneath each contributing author's name in our system to add specific details on the author's contribution. More information is available in our guide to authors.

13) Disclosure statement and competing interests: We updated our journal's competing interests policy in January 2022 and request authors to consider both actual and perceived competing interests. Please review the policy

<https://www.embopress.org/competing-interests> and update your competing interests if necessary.

14) Every published paper now includes a 'Synopsis' to further enhance discoverability. Synopses are displayed on the journal webpage and are freely accessible to all readers. They include a short stand first (maximum of 300 characters, including space) as well as 2-5 one-sentences bullet points that summarizes the paper. Please write the bullet points to summarize the key NEW findings. They should be designed to be complementary to the abstract - i.e. not repeat the same text. We encourage inclusion of key acronyms and quantitative information (maximum of 30 words / bullet point). Please use the passive voice. Please attach these in a separate file or send them by email, we will incorporate them accordingly.

Share synopsis text and image, as well as eTOC:

Please note that these would be the final versions and changes during proofing are usually not allowed

15) As part of the EMBO Publications transparent editorial process initiative (see our Editorial at <http://embomolmed.embopress.org/content/2/9/329>), Molecular Systems Biology Medicine will publish online a Review Process File (RPF) to accompany accepted manuscripts.

In the event of acceptance, this file will be published in conjunction with your paper and will include the anonymous referee reports, your point-by-point response and all pertinent correspondence relating to the manuscript. Let us know whether you agree with the publication of the RPF and as here, if you want to remove or not any figures from it prior to publication.

Molecular Systems Biology has a "scooping protection" policy, whereby similar findings that are published by others during review or revision are not a criterion for rejection. Should you decide to submit a revised version, I do ask that you get in touch after three months if you have not completed it, to update us on the status.

I look forward to receiving your revised manuscript.

Yours sincerely,

Poonam Bheda

Poonam Bheda, PhD
Scientific Editor
Molecular Systems Biology

Reviewer #1:

Summary

This study explores the transcriptional responses and associated chromatin changes in monocyte-derived macrophages of human subjects following two SARS-CoV-2 vaccination doses. Macrophages isolated after two weeks following this regime showed increased responsiveness to in vitro stimulation. In addition, patterns of H3K27 acetylation were changed at this timepoint and loci with differential H3K27ac were found to be enriched in genes related to TLR signaling pathways, regulation of cytokine production, and innate immunity GOs. Changes in acetylation were accompanied by differential gene expression in macrophages in vaccinated (2 doses) relative to unvaccinated individuals - enriched genes in similar GO categories as differential H3K27ac genes. The changes in H3K27ac patterns and gene expression are maintained long-term (to at least 12 weeks). In addition to gene expression changes, the isolated macrophages at 12 weeks maintained an increased responsiveness to some secondary innate immunity triggers.

Longer-term memory, 36 weeks after 2nd vaccination dose, showed still some trends of differential H3K27ac although no longer significant. Nevertheless, there appears to be a form of memory as a 3rd vaccination dose boosts H3K27ac changes and macrophage response to stimulation to levels beyond those of the 1st & 2nd vaccination, indicating maintenance of responsiveness.

General remarks

While the results are largely correlative, they reveal a striking degree of long-term priming of macrophages in human subjects. However, the key findings are based on a rather superficial analysis of H3K27ac and gene expression data that is only partially informative (e.g. numbers of H3K27ac differentially enriched genes). The study requires a much more elaborate and quantitative re-analysis, especially to stratify H3K27ac peaks and correlate them with transcriptomic changes and responsiveness to stimuli. In addition, the data is generally poorly annotated and described in figures and legends, and there are many plots that do not appear to add much to the overall story (see detailed points below). There are also some rather serious confounding effects resulting from using different vaccines at the later stages of the study that cast doubts on the effects of the booster vaccine at 6 months post initial priming.

Major points

H3K27ac data:

The maintenance of H3K27ac post-vaccination (up to 12 weeks following t2 and also at t5 following the 3rd dose) is one of the main findings of the paper and the authors hypothesize this to underlie epigenetic memory to SARS-CoV-2 in Macrophages. Currently, the presented results are based on gene peak counts (Fig. 2A & 4B) and a few genome browser snapshots for representative genes (Fig. 2D, 4F-G, EV2B-C & EV3F). For the former, it is unclear how the analysis was performed. The method section is vague on the methodology and the figure legends are incomplete or discrepant (e.g. Fig.2A legend: H3K27ac peaks associated with promoter sites - does it mean that the dataset excludes enhancers?). The legends should be more descriptive of what is meant by "gene peak counts".

More to the point, scoring mere differential changes in H3K27ac peaks is inadequate as it aggregates and muddles potentially informative changes in chromatin structure. For instance, a differential H3K27ac gene peak count could mean a changed enrichment (first higher than lower or visa versa) on the same genes throughout the time points, or a qualitative change (new peaks arise/former peaks disappear), re-distribution (enhancer vs promoter vs gene body accumulation), etc. By integrating the data merely as "differential changes" without regard for the direction of change and where changes are occurring relative to genes (promoters, distal elements), the authors run the risk of missing out on valuable information about how the memory of SARS-CoV-2 vaccination is propagated in macrophages.

Specifically, with regard to H3K27ac data, the authors should:

- 1) Distinguish which genes show an increased and decreased H3K27ac enrichment with analysis of their dynamics over time points as separate groups. Similar to the way gene expression data is presented in quantitative volcano plots (Figures 3A and B), H3K27ac data could be presented in this way, presenting how much acetylation is increased or reduced relative to t0 and what are the top loci that show a change.
- 2) GO analysis on genes with decreased and increased H3K27ac signal as separate groups (current claims about GO are not fully informative if based on differential H3K27ac targets).
- 3) Elaboration on whether differential H3K27ac enrichment between timepoints corresponds to gene bodies, promoters or enhancers (this can be achieved by changing parameters for peak annotation & association with nearest genes, mapping to annotated enhancers or usage of promoter-only reference).

4) Representative snapshots with both increased and lower H3K27ac should be included (current selection is based on gene sets arbitrarily selected from GO categories and with increased enrichment of H3K27ac after vaccinations).

5) Importantly, at present, the authors report that at t4, 24 weeks after t3, there are still differential effects on H3K27ac though these are not significantly different from t0. However, a more detailed analysis as requested above might reveal interesting details of some genes that maintain changes in H3K27ac. The memory of the primed state may reside in one or a few key regulators (e.g. transcription factors). Exploring this would be worthwhile.

Regarding the selection of GO terms presented in Figures 2B, C and Figure 4D, E, it is unclear what is meant by gene ratio on the x-axis. Does a higher gene ratio mean a larger difference in H3K27ac patterns between t2 and t0 (or t5 to t0)? It would help to include GO terms that are not changed to help clarify the specificity of these effects.

Also, the yellow labeled GO terms in 2B and 4D appear to be arbitrarily chosen. What is special about these GO terms compared to the others that are in grey?

Figure 2C, Are the individual gene names (red and blue) genes that are differentially expressed in t2 versus t0, or simply examples of genes that are part of the gene networks that are represented by those GO terms. Again, the description in the legend is too minimalistic.

Gene expression data:

in Fig. 3A and B: it is not clear what is the direction of the effect. For instance, in Figure 3A, "Unstimulated vs SP stimulated" implies that genes are more expressed in unstimulated samples. Similarly, for 3B what is on the left and right of the volcano plot should be more clearly indicated.

What is plotted in Figure 3D? It appears this is a subset of genes that show opposite effects upon SP stimulation in unvaccinated versus vaccinated individuals. Why is this presented and how does this relate to the genes in the Venn diagram in 3E? There is no adequate explanation of this in the text.

Confounding effects of 3rd vaccination:

Following the 3rd vaccination, an enhanced effect on H3K27ac changes and IL-1beta production is observed that are authors link to priming effects of the 1st and 2nd doses. However, this final 3rd vaccination was performed with a different vaccine formulation (Comirnaty versus Spikevax for the first two vaccinations). It is therefore possible (and perhaps likely) that the enhanced effects observed at t5 are not a consequence of priming but rather a different response of the individuals to a different vaccine. While the authors may not have access to properly controlled samples, this caveat should be clearly addressed.

Linking H3k27ac and transcription data:

The authors note an association between changes in the H3K27ac levels and the transcriptome (e.g. in the abstract and lines #197-201). To what extent these two datasets correlate is an important part of the paper. The authors should provide a clearer correlation/association analysis between H3K27ac cut&run and RNA-seq. In the current version, the association is rather incomplete and based only on the overlapping GO definitions. It would be much more informative and quantitative to address how increased H3K27ac enrichment at t2 relates to the transcriptional response of its target genes upon SP-stimulation at t2. Do all of the genes accumulating H3K27ac after 2nd vaccination have upregulated expression under SP at t2? If not, does it depend on H3K27me3 distribution (enhancer vs promoter), distance to TSS/gene body or enrichment level (threshold 'priming' level)? Transcriptional regulation & epigenetic maintenance of chromatin are multifactorial processes - such correlation would show to what extent H3K27ac is predictive in gene expression outcome and would increase the impact of claims made in this paper.

In the discussion, the authors should elaborate on the paradox of how short-lived macrophages can maintain a long-term innate immune memory. Elaborate on the section in lines #255-259 to include examples of bone marrow HSCs reprogramming and how this may explain long-term memory as observed in the experiments (t5, Fig.4) (the caveat of Comirnaty versus Spikevax notwithstanding).

Also, a more general discussion on how the authors envision how memory is maintained is missing. Continued gene expression? Propagation of mitotically heritable chromatin structure?

Minor points

- Generally, the authors refer to Dataset EV1 through EV10 which are simple lists of genes or GO terms that are not annotated or explained, and one is left to decipher what these Excel sheets represent. For instance, if this reviewer is correct, Dataset EV5 lists the GO terms of genes associated with changes in H3K27Ac at time point t3 relative to t0. It would help to have a list of such captions, specifying what each of the datasets presents, in the main manuscript.

- Fig 1F-I & EV1A-D: It is not clear what the grey and red dots indicate. From the text, one can infer that these correspond to

supernatants from isolated macrophages of unvaccinated and vaccinated patients, respectively but this needs to be more explicit.

- Fig. EV2A, EV3E: heatmap scale should be described. Does it show signal strength as read count (CPM)?
- Fig.2B-C: It is unclear what the grey and yellow color indicates in Fig.2B - it is described only in the legend for Fig 2C. Also in 2C, 5 GO terms are elaborated based on being yellow in 2B but there are 6 GO terms highlighted in 2B. The term "Toll-like receptor signaling pathway" is not elaborated in 2C.
- Long sentences such as #167-170 - it would be better for clarity to rephrase them and split into 2-3 sentences.
- Fig. 3C: Indicate that these DEGs correspond to samples isolated at t2
- Line 229: Please clarify what is meant by PRR genes with histone marks. It is unclear what histone marks are meant in the sentence.
- For clarity, it would be good to see DEG criteria also in the main text (methods: DEGs are scored here as genes with $\log_2 > 1$ and FDR 0.05)
- Methods: cut&run data analysis - the method to score nearest genes to H3K27ac peaks should be specified along with parameters used.

Reviewer #3:

How exactly mRNA vaccines activate different layers of the innate and adaptive immune system remains not fully understood. The lab has several papers using the model and demonstrates that both SARS-CoV-2 infection and mRNA vaccination prime human monocyte-derived macrophages for potent secretion of proinflammatory cytokines following re-stimulation with the SARS-CoV-2 SP ex vivo. This manuscript by Simonis and colleagues is a follow-up study that further studies how SARS-CoV-2 mRNA vaccination induces IL1b, the data showed that mRNA vaccination leads to profound alterations in the histone H3 lysine 27 acetylation (H3K27ac) profile of human monocytes-derived macrophages, they further demonstrate that two consecutive vaccinations were required for substantial pattern changes of H3K27ac. Overall, the study is well-planned and executed, and the conclusions are justified. My main criticism of the work is it lacks a bit of novelty, and that the authors do not go deeply into the mechanism.

In Figure 1, the authors showed that several cytokines and chemokines (TNF- α , IL-36, CCL3, CCL20, CCL4, CXCL1) were significantly elevated, while some were not affected, KINK-1 (kinase inhibitor of NF- κ B) was used to demonstrate that NF- κ B plays an important role, it is interesting to know that why the author focuses on the NF- κ B not other transcriptional factors, the authors were recommended to test whether other transcriptional factors were involved. In addition, it is interesting to know why there are various sample numbers in T1 or T2 group treated with ssRNA, Zymosan, or Pam3.

In Figure 2, CUT&RUN was performed to determine genomic distribution patterns of H3K27ac at three post-vaccination time points. It is interesting to know which and how the histone acetylases are responsible for the modification and this will deepen the mechanism.

Rebuttal for Simonis et al.: Persistent epigenetic memory of SARS-CoV-2 mRNA vaccination in monocyte-derived macrophages

We would like to thank the reviewers for their interest in our work and the constructive points raised that clearly improved the manuscript. We have addressed those in a point-by-point manner (reviewer comments in grey and italics). Our response is shown in blue font. Changes in the manuscript are highlighted in red.

Reviewer #1:

Summary

This study explores the transcriptional responses and associated chromatin changes in monocyte-derived macrophages of human subjects following two SARS-CoV-2 vaccination doses. Macrophages isolated after two weeks following this regime showed increased responsiveness to in vitro stimulation. In addition, patterns of H3K27 acetylation were changed at this timepoint and loci with differential H3K27ac were found to be enriched in genes related to TLR signaling pathways, regulation of cytokine production, and innate immunity GOs. Changes in acetylation were accompanied by differential gene expression in macrophages in vaccinated (2 doses) relative to unvaccinated individuals - enriched genes in similar GO categories as differential H3K27ac genes. The changes in H3K27ac patterns and gene expression are maintained long-term (to at least 12 weeks). In addition to gene expression changes, the isolated macrophages at 12 weeks maintained an increased responsiveness to some secondary innate immunity triggers. Longer-term memory, 36 weeks after 2nd vaccination dose, showed still some trends of differential H3K27ac although no longer significant. Nevertheless, there appears to be a form of memory as a 3rd vaccination dose boosts H3K27ac changes and macrophage response to stimulation to levels beyond those of the 1st & 2nd vaccination, indicating maintenance of responsiveness.

General remarks

While the results are largely correlative, they reveal a striking degree of long-term priming of macrophages in human subjects. However, the key findings are based on a rather superficial analysis of H3K27ac and gene expression data that is only partially informative (e.g. numbers of H3K27ac differentially enriched genes). The study requires a much more elaborate and quantitative re-analysis, especially to stratify H3K27ac peaks and correlate them with transcriptomic changes and responsiveness to stimuli. In addition, the data is generally poorly annotated and described in figures and legends, and there are many plots that do not appear to add much to the overall story (see detailed points below). There are

also some rather serious confounding effects resulting from using different vaccines at the later stages of the study that cast doubts on the effects of the booster vaccine at 6 months post initial priming.

Response 1:

We thank the reviewer for considering this work an important contribution to the effect of vaccination on the long-term memory of immune cells and agree that a deeper analysis of the CUT&RUN is required and would provide more insights into the pathways impacted. Our revised manuscript provides a much deeper evaluation of CUT&RUN data strongly supporting our main hypothesis (**Line 187 – 268**).

Major points

H3K27ac data:

The maintenance of H3K27ac post-vaccination (up to 12 weeks following t2 and also at t5 following the 3rd dose) is one of the main findings of the paper and the authors hypothesize this to underlie epigenetic memory to SARS-CoV-2 in Macrophages. Currently, the presented results are based on gene peak counts (Fig. 2A & 4B) and a few genome browser snapshots for representative genes (Fig. 2D, 4F-G, EV2B-C & EV3F). For the former, it is unclear how the analysis was performed. The method section is vague on the methodology and the figure legends are incomplete or discrepant (e.g. Fig.2A legend: H3K27ac peaks associated with promoter sites - does it mean that the dataset excludes enhancers?). The legends should be more descriptive of what is meant by "gene peak counts".

Response 2:

We agree with the feedback and have added more details to the figure legends and Materials and Methods section to clarify what is presented (e.g., **Line 494 – 559**). Initially, we showed only the number of peaks that changed significantly for each patient/time point across all patients, as a straightforward way to visualize overall changes over the course of the time. We have now refined our approach to consider continuous H3K27ac signals in the identified peaks, quantifying the mean fold-changes (log2 ratios) of H3K27ac sequencing read coverages at these peaks across patients at each time point (t1, t2, t3, t4, t5), relative to unvaccinated probands (t0). Additionally, we have generated a comprehensive, step-by-step script (**Dataset EV7**) as a supporting document, detailing our data transformation and normalization strategy.

More to the point, scoring mere differential changes in H3K27ac peaks is inadequate as it aggregates and muddles potentially informative changes in chromatin structure. For instance, a differential H3K27ac gene peak count could mean a changed enrichment (first higher than lower or visa versa) on the same genes throughout the time points, or a

qualitative change (new peaks arise/former peaks disappear), re-distribution (enhancer vs promoter vs gene body accumulation), etc. By integrating the data merely as "differential changes" without regard for the direction of change and where changes are occurring relative to genes (promoters, distal elements), the authors run the risk of missing out on valuable information about how the memory of SARS-CoV-2 vaccination is propagated in macrophages.

Specifically, with regard to H3K27ac data, the authors should:

1) Distinguish which genes show an increased and decreased H3K27ac enrichment with analysis of their dynamics over time points as separate groups. Similar to the way gene expression data is presented in quantitative volcano plots (Figures 3A and B), H3K27ac data could be presented in this way, presenting how much acetylation is increased or reduced relative to t0 and what are the top loci that show a change.

Response 3:

We agree and now added violin distributions of H3K27ac fold changes in H3K27ac peaks located in three categories, a) gene promoters (-1kb to +250 bp from TSS), b) gene bodies (+251bp from TSS until TES) and in c) distal regions (100kb - 1kb away from TSS and TES) (please see **Fig. 3B-D**). In addition, we followed the reviewer's advice to quantify mean fold changes in peaks associated to genes that fall into the category a) promoter, b) gene body and c) distal regions and plotted changes per gene across time points (please see methods and script [**Dataset EV7**] for the details of the analysis). Importantly, we performed unsupervised clustering of H3K27ac fold changes across time points for genes in each category (promoters, gene body, distal regions) to identify clusters of genes exhibiting similar alterations in H3K27ac fold changes over time. Our analysis revealed a multidimensional granularity in H3K27ac signal changes over time (**Fig 4A-C**). Notably, the largest gene cluster (~8,000 genes; see **Fig 4A and Fig 4D**) exhibited promoters with significantly higher H3K27ac fold changes in t4 compared to t1 ($p < 0.0001$) (**Fig 4E**). In contrast, the largest clusters associated with H3K27ac peaks in gene bodies and distal regions showed significantly lower fold changes in t4 versus t1 (**Fig 4F and G**). This suggests that a substantial number of genes have promoters exhibiting a similar acquired, persistent epigenetic memory, whereas gene bodies and distal regions lack this feature (**Fig 4E-G**).

2) GO analysis on genes with decreased and increased H3K27ac signal as separate groups (current claims about GO are not fully informative if based on differential H3K27ac targets).

Response 4:

We performed unsupervised association of all gene clusters, including promoters, gene bodies and distal regions, against immune GO terms. Genes part of cluster 1 related to the

promoter category exhibited a significant increase in the H3K27ac fold changes at t4 relative to t1 (**Fig. 4A**) and significantly associated with immune terms, please see **Dataset EV4 and Fig 5A** .

3) *Elaboration on whether differential H3K27ac enrichment between timepoints corresponds to gene bodies, promoters or enhancers (this can be achieved by changing parameters for peak annotation & association with nearest genes, mapping to annotated enhancers or usage of promoter-only reference).*

Response 5:

We are grateful for the reviewers' suggestions. In response, we differentiated between H3K27ac peak enrichments in promoters, gene bodies, and distal regions. Please refer to Response 3 for a detailed explanation.

4) *Representative snapshots with both increased and lower H3K27ac should be included (current selection is based on gene sets arbitrarily selected from GO categories and with increased enrichment of H3K27ac after vaccinations).*

Response 6:

We have included H3K27ac coverage data along with genome browser snapshots of gene promoters belonging to cluster 1 (including IL1B and IL18), which suggest an overall increase in H3K27ac at time points t1 to t5 relative to t0. Additionally, we present gene promoters from cluster 14 (e.g., SLC1A2 and CES3) that are not associated with immune-related terms and do not show a persistent increase in H3K27ac (t4 vs. t0). For transparency, we also display data for the other time points, as shown in **Response Figs. 1 and 2**.

Response Figure 1: H3K27ac coverage shown as log2 fold change across time points (t1-t5) relative to t0 for Cluster 1 (**A**) and Cluster 14 (**B**). Timepoint t4 is highlighted.

Response Figure 2: Genome browser snapshots of promoters from genes listed in cluster 1 and cluster 14 displaying differential H3K27ac fold changes across time points (t1-t5) relative to t0. Red indicates increased H3K27ac relative to t0 and blue decreased H3K27ac relative to t0.

5) Importantly, at present, the authors report that at t4, 24 weeks after t3, there are still differential effects on H3K27ac though these are not significantly different from t0. However, a more detailed analysis as requested above might reveal interesting details of some genes that maintain changes in H3K27ac. The memory of the primed state may reside in one or a few key regulators (e.g. transcription factors). Exploring this would be worthwhile.

Response 7:

We show now in **Figure 3E** that promoters specifically and significantly exhibit an increased H3K27ac signal at t4/t0 relative to gene bodies and distal regions. We also show in **Figure 4C-D** that promoters show a significantly increased t4/t0 H3K27ac signal relative to t1/t0, while a significantly decreased (t4/t0) H3K27ac signal in gene bodies and distal regions relative to t1/t0. These results demonstrate that, independent of t0, promoters show a significantly higher H3K27ac signal in t4 relative to t1.

Regarding the selection of GO terms presented in Figures 2B, C and Figure 4D, E, it is unclear what is meant by gene ratio on the x-axis. Does a higher gene ratio mean a larger difference in H3K27ac patterns between t2 and t0 (or t5 to t0)?

It would help to include GO terms that are not changed to help clarify the specificity of these effects. Also, the yellow labeled GO terms in 2B and 4D appear to be arbitrarily chosen. What is special about these GO terms compared to the others that are in grey?

Response 8:

The epigenetic analysis has been revised substantially. **Figures 2B and 2C**, as well as **Figures 4D and 4E**, have been removed. Novel data address this and other points of critique made by the reviewer.

Figure 2C, Are the individual gene names (red and blue) genes that are differentially expressed in t2 versus t0, or simply examples of genes that are part of the gene networks that are represented by those GO terms. Again, the description in the legend is too minimalistic.

Response 9:

See response 8.

Gene expression data:

in Fig. 3A and B: it is not clear what is the direction of the effect. For instance, in Figure 3A, "Unstimulated vs SP stimulated" implies that genes are more expressed in unstimulated samples. Similarly, for 3B what is on the left and right of the volcano plot should be more clearly indicated.

Response 10:

The figure legend has been revised for improved clarity (now **Fig 2A and B**) (**Line 798 – 804**): "Volcano plot showing differentially expressed genes (grey dots) in SP-stimulated macrophages (n = 6) compared to unstimulated macrophages (n = 6) at t0 for unvaccinated (A) and vaccinated (t2) (B) individuals. Negative log₁₀ adjusted p-values are plotted against the log₂ fold-change. Downregulated genes are depicted with negative log₂ fold-change values on the left side of the plot, whereas upregulated genes are represented by positive log₂ fold-change values on the right side. Dotted lines indicate log₂ fold-change ± 1 and -log₁₀ adjusted p-values of 1."

What is plotted in Figure 3D? It appears this is a subset of genes that show opposite effects upon SP stimulation in unvaccinated versus vaccinated individuals. Why is this presented and how does this relate to the genes in the Venn diagram in 3E? There is no adequate explanation of this in the text.

Response 11:

In the revised version we explain the data (now **Fig 2C and D**) in more detail in the main text (**Line 153 – 161**):

"When comparing the transcriptome of SP-stimulated cells of both groups (unvaccinated versus vaccinated), the two groups were clearly distinguishable based on their gene expression profiles (Fig 2C). Gene expression analysis in macrophages following SP

stimulation, compared to unstimulated cells, revealed 2,519 genes that were differentially expressed (defined by a log₂ fold-change of ± 1 and an adjusted p-value < 0.05) exclusively in macrophages derived from vaccinated individuals. Additionally, 268 differentially expressed genes (DEGs) were shared between both groups, while only 50 DEGs were specific to macrophages from unvaccinated individuals (Fig 2D)."

Confounding effects of 3rd vaccination:

Following the 3rd vaccination, an enhanced effect on H3K27ac changes and IL-1beta production is observed that are authors link to priming effects of the 1st and 2nd doses. However, this final 3rd vaccination was performed with a different vaccine formulation (Comirnaty versus Spikevax for the first two vaccinations). It is therefore possible (and perhaps likely) that the enhanced effects observed at t5 are not a consequence of priming but rather a different response of the individuals to a different vaccine. While the authors may not have access to properly controlled samples, this caveat should be clearly addressed.

Response 12:

We now discuss this issue in the revised manuscript. However, the fact that there was a switch of vaccine provider has little impact on the fact that we detected long term epigenetic modifications of promoters at t4, just prior to the third vaccination. This finding should be completely independent of the vaccine construct used and there was no switch affecting these data. Regarding the third vaccine applied, we do not believe that use of Comirnaty is the reason for such potent innate immune responses after booster vaccination. Most studies focussing on effectiveness and correlates of immune protection (mostly adaptive immune system) showed almost identical results. Importantly, the cytokine secretion data were performed with more vaccinated individuals. Here we had included some vaccinees that received **Spikevax** first and then Comirnaty. In these individuals we observed the same rather weak response after the first vaccination and a much stronger response after the second vaccination. This corroborates many findings showing that both vaccines are comparable with regard to the human immune response following vaccination. For the epigenetic evaluation we wanted to avoid differential use of vaccines at a single time-point.

Linking H3k27ac and transcription data:

The authors note an association between changes in the H3K27ac levels and the transcriptome (e.g. in the abstract and lines #197-201). To what extent these two datasets correlate is an important part of the paper. The authors should provide a clearer correlation/association analysis between H3K27ac cut&run and RNA-seq. In the current version, the association is rather incomplete and based only on the overlapping GO

definitions. It would be much more informative and quantitative to address how increased H3K27ac enrichment at t2 relates to the transcriptional response of its target genes upon SP-stimulation at t2. Do all of the genes accumulating H3K27ac after 2nd vaccination have upregulated expression under SP at t2? If not, does it depend on H3K27me3 distribution (enhancer vs promoter), distance to TSS/gene body or enrichment level (threshold 'priming' level)? Transcriptional regulation & epigenetic maintenance of chromatin are multifactorial processes - such correlation would show to what extent H3K27ac is predictive in gene expression outcome and would increase the impact of claims made in this paper.

Response 13:

We agree with the reviewer that this could be an interesting point for the manuscript. We have discovered that epigenetic memory by means of H3K27ac is established after the second vaccination (t2) and persists (t4) at specific gene promoters, such as immune genes. To address whether the promoters of genes, which show increased expression levels in response to SP-stimulation, are persistently marked by H3K27ac, we quantified the fold change of H3K27ac read coverages at the respective time points (t1-t5) relative to t0 (unvaccinated) in H3K27ac peaks that overlap with promoters of genes that exhibit increased transcriptional activities (SP-stimulated genes). We found that H3K27ac is specifically established at t2 and persists in t4, both relative to t0, in these SP-stimulated gene promoters (**Fig. EV2A**). Importantly, we further revealed that the overlap between the H3K27ac peaks found in the SP-stimulated promoters (576) show a highly significant overlap with the H3K27ac peaks in promoters of Cluster1 Immune-associated genes (333), please see figure **EV2B**.

In the discussion, the authors should elaborate on the paradox of how short-lived macrophages can maintain a long-term innate immune memory. Elaborate on the section in lines #255-259 to include examples of bone marrow HSCs reprogramming and how this may explain long-term memory as observed in the experiments (t5, Fig.4) (the caveat of Comirnaty versus Spikevax notwithstanding).

Response 13:

There now is novel data revealing epigenetic reprogramming of monocytes following SARS-CoV-2 infection which is linked to reprogramming of stem cells. These data are cited in the discussion (Cheong et al. Cell 2023; PMID: 37597510). It is thus very likely that the respective mRNA vaccine has similar impact on bone marrow cells.

Also, a more general discussion on how the authors envision how memory is maintained is missing. Continued gene expression? Propagation of mitotically heritable chromatin structure?

Response 14:

We now discuss the possibility that G-quadruplex DNA sequences at the promoters of vaccine-induced genes (please see **Fig. 3F and 5B**) and promote long-term activity of these genes and persistence of H3K27ac as transcriptional active mark at promoters. G-quadruplex structures are single-stranded phenomena in transcribed regions and their active removal lead to an increase in nucleosome-density, which is necessary to permanently shut-down transcription.

Minor points

- Generally, the authors refer to Dataset EV1 through EV10 which are simple lists of genes or GO terms that are not annotated or explained, and one is left to decipher what these Excel sheets represent. For instance, if this reviewer is correct, Dataset EV5 lists the GO terms of genes associated with changes in H3K27Ac at time point t3 relative to t0. It would help to have a list of such captions, specifying what each of the datasets presents, in the main manuscript.

Response 15:

In the revised version we have now made sure to upload well-labeled supplementary data and corresponding descriptions in the main body of the text.

- Fig 1F-I & EV1A-D: It is not clear what the grey and red dots indicate. From the text, one can infer that these correspond to supernatants from isolated macrophages of unvaccinated and vaccinated patients, respectively but this needs to be more explicit.

Response 16:

In the revised version we added this information in the figure legends: **(Line 789 – 791)** “...*Monocyte-derived macrophages from unvaccinated (t0) (grey) and vaccinated individuals (t2) (red) were generated, stimulated with SP (t0: dark grey dots; t2: dark red dots) or left unstimulated (t0: light grey dots; t2: light red dots) as described in (C).*”

Expanded view data: “*Monocytes were isolated by CD14+ selection from blood samples from unvaccinated (t0) (grey) and vaccinated individuals (t2) (red) and seeded and incubated in the presence of M-CSF for 5 days. Differentiated macrophages were stimulated with SP (t0: dark grey dots; t2: dark red dots) or left unstimulated (t0: light grey dots; t2: light red dots).*”

- Fig. EV2A, EV3E: heatmap scale should be described. Does it show signal strength as read count (CPM)?

Response 17:

The graphs were removed in current version of the manuscript due to substantial changes of the epigenetic data.

- Fig.2B-C: It is unclear what the grey and yellow color indicates in Fig.2B - it is described only in the legend for Fig 2C. Also in 2C, 5 GO terms are elaborated based on being yellow in 2B but there are 6 GO terms highlighted in 2B. The term "Toll-like receptor signaling pathway" is not elaborated in 2C.

Response 18:

The epigenetic part has been revised substantially. Figures 2B and 2C have been removed.

- Long sentences such as #167-170 - it would be better for clarity to rephrase them and split into 2-3 sentences.

Response 19:

The revised manuscript has undergone substantial changes also in the format. To improve clarity, we aimed to avoid long sentences in this version.

- Fig. 3C: Indicate that these DEGs correspond to samples isolated at t2

Response 20:

In the revised version we added more information in the figure legend (now **Fig 2C**). *“Venn diagram showing number of differentially expressed genes (DEGs) in monocyte-derived macrophages of unvaccinated (blue) (n = 6) and vaccinated (red) (n = 6) individuals upon SP stimulation compared to unstimulated cells. The red color represents the number of DEGs unique to vaccinated individuals, while the blue color indicates DEGs unique to unvaccinated individuals. The yellow color highlights the overlapping DEGs between the two groups. Circle sizes of the venn diagram correspond to the number of genes.” (Line 804 – 811).*

- Line 229: Please clarify what is meant by PRR genes with histone marks. It is unclear what histone marks are meant in the sentence.

Response 21:

The epigenetic section, including this sentence, has been substantially revised.

- For clarity, it would be good to see DEG criteria also in the main text (methods: DEGs are scored here as genes with $\log_2 > 1$ and FDR 0.05)

Response 22:

In the revised version we added this information to the main text (**Line 157 – 158**).

- Methods: cut&run data analysis - the method to score nearest genes to H3K27ac peaks should be specified along with parameters used.

Response 23:

In the revised version we extensively increased the description of bioinformatics analyses. With respect to the connection of H3K27ac peaks to the nearest gene feature (promoter, gene body, distal regions), we refer to our method section (CUT&RUN data analysis).

Reviewer #3:

How exactly mRNA vaccines activate different layers of the innate and adaptive immune system remains not fully understood. The lab has several papers using the model and demonstrates that both SARS-CoV-2 infection and mRNA vaccination prime human monocyte-derived macrophages for potent secretion of proinflammatory cytokines following re-stimulation with the SARS-CoV-2 SP ex vivo. This manuscript by Simonis and colleagues is a follow-up study that further studies how SARS-CoV-2 mRNA vaccination induces IL1b, the data showed that mRNA vaccination leads to profound alterations in the histone H3 lysine 27 acetylation (H3K27ac) profile of human monocytes-derived macrophages, they further demonstrate that two consecutive vaccinations were required for substantial pattern changes of H3K27ac. Overall, the study is well-planned and executed, and the conclusions are justified. My main criticism of the work is it lacks a bit of novelty, and that the authors do not go deeply into the mechanism.

Response 24:

We would like to express our appreciation to reviewer #3 for dedicating time to evaluate our manuscript.

With regard to novelty of our manuscript, we would like to mention, that, at the time of submission, there was only one other publication addressing epigenetic memory in the context of SARS-CoV-2 mRNA vaccination. This work is technically distinct and the length of the observation period of vaccinated individuals, which is a key parameter in epigenetics research, was significantly shorter. The paper by Yamaguchi et al (PMID: 36282593) was cited.

Yamaguchi et al. used ATAC-seq experiments of monocytes to show that epigenetic memory is short lived in these cells. However, the latest time-point of observation was day 49 after the first vaccination. Our study includes epigenetic data of the third vaccination six months after first and second vaccination which allowed us to claim that epigenetic memory is indeed long lasting and required for a rapid immune response after priming vaccines were injected. This is a novel and highly interesting finding in the context of mRNA vaccines. In addition, we were using the more advanced CUT&RUN technique to generate high quality epigenetic data. In addition, we established a non-canonical role of H3K27ac to establish

epigenetic memory at promoters containing G-quadruplex DNA sequences, which has to our knowledge not been reported previously.

To the best of our knowledge, there is no other publication investigating epigenetic memory following mRNA vaccination with such long observation periods with 2 booster vaccinations.

In Figure 1, the authors showed that several cytokines and chemokines (TNF- α , IL-36, CCL3, CCL20, CCL4, CXCL1) were significantly elevated, while some were not affected, KINK-1 (kinase inhibitor of NF- κ B) was used to demonstrate that NF- κ B plays an important role, it is interesting to know that why the author focuses on the NF- κ B not other transcriptional factors, the authors were recommended to test whether other transcriptional factors were involved. In addition, it is interesting to know why there are various sample numbers in T1 or T2 group treated with ssRNA, Zymosan, or Pam3.

Response 25:

Our manuscript is a systems biology paper providing for the first time, insight into the transcriptomic and epigenetic landscape of a specific blood compartment following mRNA vaccination over an observation period of several months. The cytokine data were presented to link omics/systems data to an immunological outcome and to confirm that the vaccinated individuals of our cohort react as expected. These systems biology data are following up on recently published data looking into the exact molecular signalling mechanism in monocytes/macrophages of vaccinated individuals (Theobald et al. EMBO Mol Med 2022; PMID: 35785445). In this manuscript, we were able to reveal that the spleen tyrosine kinase SYK is required for cytokine secretion in macrophages following vaccination. SYK activation and phosphorylation was upstream of NF- κ B which was confirmed using multiple, technically distinct methods (PMID: 35785445). The manuscript showing these data was cited several times in our novel systems biology manuscript and we do not see the possibility to add additional mechanistic data which were not yet provided in already published work. SYK is a relatively global regulator of cytokine secretion and we were now able to show that vaccination affects a large series of key pro-inflammatory cytokines and chemokines following ex-vivo stimulation of macrophages. Small molecules such as KINK-1 were rather used as controls to confirm that signaling in these macrophages can be shut down and that we are not measuring an artifact of lysed cells. It is not fully clear why a small set of chemokines failed to respond with increased secretion. This may well be a technical issue. For these “non-responding” cytokines, signaling pathways are not well described and deciphering involved kinases would be out of scope of this work.

A key purpose of our work was to confirm that epigenetic modifications correlate with the interesting observation of short-lived monocytes remaining in an activated state over several months which allows for potent cytokine release following the third booster vaccination. To

this end, it is highly interesting that we found significant epigenetic alterations (H3K27ac peaks) of the gene coding for SYK (**Fig. EV3B**). Thus, our two manuscripts strongly build on each other and the new data provide an important missing link to our knowledge of long-lived epigenetic and mechanistic changes of short-lived immune cells.

In Figure 2, CUT&RUN was performed to determine genomic distribution patterns of H3K27ac at three post-vaccination time points. It is interesting to know which and how the histone acetylases are responsible for the modification and this will deepen the mechanism.

Response 26:

We thank the reviewer for raising this interesting point. We agree that further in-depth analysis of the molecular mechanisms would indeed be valuable. However, due to the limited availability of human monocytes from donors vaccinated at various time points (including individuals without vaccination), a mechanistic follow-up of the CUT&RUN data is, in our view, technically unfeasible and beyond the scope of our current study.

Nevertheless, the literature provides substantial evidence regarding the primary histone acetyltransferase responsible for the deposition of H3K27ac, which is CBP/p300 (e.g., PMIDs: 19700617, 30110629, 36215692). Additionally, there are numerous publications detailing the role of CBP/p300 in the cellular response to inflammatory signaling.

Given these technical limitations and the distinct focus of our manuscript, which does not aim to unravel underlying molecular mechanisms, we hope the reviewer will consider accepting the manuscript in its current form without further mechanistic studies, which we believe are beyond the scope of the present work.

21st Jan 2025

Manuscript Number: MSB-2023-11820R

Title: Persistent epigenetic memory of SARS-CoV-2 mRNA vaccination in monocyte-derived macrophages

Author: Alexander Simonis

Sebastian Theobald

Anna Koch

Ram Mummadavarapu

Julie Mudler

Andromachi Pouikli

Ulrike Göbel

Richard Acton

Sandra Winter

Alexandra Albus

Dmitriy Holzmann

Marie-Christine Albert

Michael Hallek

Henning Walczak

Manuel Koch

Peter Tessarz

Robert Hänsel-Hertsch

Jan Rybniker

Dear Jan,

Thank you for submitting your revised manuscript. We have now heard back from the two reviewers who agreed to evaluate your study. As you will see from the comments below, Reviewer#3 is satisfied with the revisions. However, while Reviewer #4 (who co-reviewed the initial submission with Reviewer #1 in the previous round) acknowledges the significant improvements, they still think there are several important issues with the H3K27ac data analysis.

In principle, our editorial policy allows for only one round of major revision. However, since H3K27ac data are central to supporting the major claims of the study, we would ask you to fully address Reviewer #4's concerns in this regard in an exceptional second round of review. Please feel free to contact me in case you would like to discuss in further detail any of the issues raised by the reviewer.

On a more editorial level, please address the following:

1. Please provide up to five keywords.
2. Remove the Authors' contribution section from the manuscript file.
3. EV figures need to be uploaded as separate production quality figure files and their legends should be placed in the manuscript file, after the main figure legends.
4. Author name 'Thomas Ulas' is missing from the submission system- this issue needs to be corrected.
5. Please remove "data not shown" on p23. As per our guidelines on "Unpublished Data", the journal does not permit citation of "Data not shown". All data referred to in the paper should be displayed in the main or Expanded View figures.
6. Source data: please fill out the SourceData checklist (sent by our source data coordinator on 25.08.2023) and return it when you submit the revised manuscript and provide the requested source data files along with your resubmission.
7. Data and code availability:
 - Please rename the section to "Data availability".
 - Please provide a specific URL for EGAS50000000341 dataset and make sure the dataset will be made publicly available upon the acceptance of the manuscript.
 - Remove the metadata repository information for reviewers.
 - Computer code should be deposited to an appropriate public database (e.g, GitHub).
8. EV Datasets: the sheet title for Datasets EV2, EV3, EV5 and EV6 is "Tabelle1" instead of "Dataset EV#", which need to be corrected.

9. Callouts: all instances of "Supporting Code" and "Supporting Information" should be updated to the correct callouts.
10. Funding information: We noticed discrepancies between the acknowledged funders in the submission system and those listed in the manuscript file. Specifically, the following funders are acknowledged in the manuscript but are missing in the submission system: the Career Advancement Groups Program of the Center of Molecular Medicine Cologne, Faculty of Medicine and University Hospital of Cologne, University of Cologne; CMMC - B10; Imhoff-Stiftung and Cologne Fortune; Max Planck Society; Ministry of Culture and Science of the State of Northrhine Westphalia. Please ensure that the information entered in the submission system is consistent with the manuscript file.
11. "Competing interests" should be renamed to "Disclosure Statement and Competing Interests".
12. BioRender should be acknowledged at the end of the Methods section in the following way: "Graphics: (some of the... OR Figure #... OR synopsis) Graphics were created with BioRender.com."
13. "Literature" should be renamed to "References".
14. All Materials and Methods need to be described in the main text using our 'Structured Methods' format. According to this format, the Methods section includes a Reagents and Tools Table (listing key reagents, experimental models, software and relevant equipment and including their sources and relevant identifiers) followed by a Methods and Protocols section describing the methods, ideally using a step-by-step protocol format. The aim is to facilitate adoption of the methodologies across labs.

Please download and fill our Reagents and Tools Table template (.docx), which you can find in our author guidelines: <https://www.embopress.org/page/journal/17444292/authorguide#structuredmethods>.

An example of a Method paper with Structured Methods can be found here: <https://www.embopress.org/doi/10.15252/msb.20178071>.

15. Please provide a "standfirst text" summarizing the study in one or two sentences (approximately 250 characters, including space), three to four "bullet points" highlighting the main findings and a "synopsis image" (550px width and 400-600 px height, PNG format) to highlight the paper on our homepage. Please note that the current synopsis image is too large.

16. Please fix the following issues in figure legends:

- Please note that the exact p values are not provided in the legends of figures 1C, F, G; 3E; 4E-G.
- Please indicate the statistical test used for data analysis in the legends of figures 2A, B, E, F.
- Please note that the box plots need to be defined in terms of minima, maxima, centre, bounds of box and whiskers, and percentile in the legends of figures EV1 A, F, G.
- Please note that information related to n is missing in the legends of figures 3B, C, D, E; 4E-G; EV1 A-E; EV4.

When you resubmit your manuscript, please download our CHECKLIST (<https://bit.ly/EMBOPressAuthorChecklist>) and include the completed form in your submission.

Please note that the Author Checklist will be published alongside the paper as part of the transparent process (<https://www.embopress.org/page/journal/17444292/authorguide#transparentprocess>).

If you feel you can satisfactorily deal with these points and those listed by the referees, you may wish to submit a revised version of your manuscript. Please attach a covering letter giving details of the way in which you have handled each of the points raised by the referees. A revised manuscript will be once again subject to review and you probably understand that we can give you no guarantee at this stage that the eventual outcome will be favorable.

I look forward to receiving a revised manuscript soon.

Kind regards,
Jingyi

Jingyi Hou, PhD
Senior Editor
Molecular Systems Biology

We realize that it is difficult to revise to a specific deadline. In the interest of protecting the conceptual advance provided by the work, we recommend a revision within 3 months (21st Apr 2025). Please discuss the revision progress ahead of this time with the editor if you require more time to complete the revisions. Use the link below to submit your revision:

IMPORTANT: When you send your revision, we will require the following items:

1. the manuscript text in LaTeX, RTF or MS Word format
2. a letter with a detailed description of the changes made in response to the referees. Please specify clearly the exact places in the text (pages and paragraphs) where each change has been made in response to each specific comment given
3. three to four 'bullet points' highlighting the main findings of your study
4. a short 'blurb' text summarizing in two sentences the study (max. 250 characters)
5. a 'thumbnail image' (550px width and max 400px height, Illustrator, PowerPoint or jpeg format), which can be used as 'visual title' for the synopsis section of your paper.

6. Please include an author contributions statement after the Acknowledgements section (see

<https://www.embopress.org/page/journal/17444292/authorguide>)

7. Please complete the CHECKLIST available at (<https://bit.ly/EMBOPressAuthorChecklist>).

Please note that the Author Checklist will be published alongside the paper as part of the transparent process

(<https://www.embopress.org/page/journal/17444292/authorguide#transparentprocess>).

See also figure legend guidelines: <https://www.embopress.org/page/journal/17444292/authorguide#figureformat>

9. Please note that corresponding authors are required to supply an ORCID ID for their name upon submission of a revised manuscript (EMBO Press signed a joint statement to encourage ORCID adoption).

(<https://www.embopress.org/page/journal/17444292/authorguide#editorialprocess>)

Currently, our records indicate that the ORCID for your account is 0000-0001-8351-2690.

Link Not Available

11. Include a Reagents and Tools Table as part of the Methods section, which can be downloaded from our author guidelines (<https://www.embopress.org/page/journal/17444292/authorguide#structuredmethods>)

*** PLEASE NOTE *** As part of the EMBO Press transparent editorial process initiative (see our Editorial at <https://dx.doi.org/10.1038/msb.2010.72>), Molecular Systems Biology publishes online a Review Process File with each accepted manuscripts. This file will be published in conjunction with your paper and will include the anonymous referee reports, your point-by-point response and all pertinent correspondence relating to the manuscript. If you do NOT want this File to be published, please inform the editorial office at msb@embo.org within 14 days upon receipt of the present letter.

Reviewer #3:

All of my concerns have been addressed

Reviewer #4:

Summary

Modified from summary in revision round 1. This study explores the transcriptional responses and associated chromatin changes in monocyte-derived macrophages of human subjects following two SARS-CoV-2 vaccination doses. Monocytes-derived macrophages isolated after 2 weeks following this regime showed increased responsiveness to in vitro stimulation, which remained high (significant only for ssRNA-stimulation) up to 10 weeks following second vaccination. High responsiveness

is achieved after two vaccinations, suggesting prime-boost effect. The authors related changes in responsiveness with upregulated transcriptional activity of immune response genes, increased global enrichment of H3K27ac in the chromatin and enhanced cytokine production (all in stimulated macrophages from individuals with 2 vaccination doses (t2) vs unvaccinated (t0)). In addition, the authors showed that H3K27ac changes are long-lasting - its enrichment is further increased at the 3rd vaccination dose (t5; applied 6 months after the 2nd) and is retained long-term ~6 months after the 2nd dose (t4) specifically at gene promoters. The authors clustered H3K27ac data to identify gene group which belong to GOs for immune response genes, contain G4-quadruplexes in the promoters and show similar transcriptional profile to transcriptionally activated genes after 2nd vaccination (t2). Overall, this study suggests a long-term memory in monocyte-derived macrophages to SARS-CoV-2 vaccinations, correlated with increased responsiveness to immune triggers, increased stimulated cytokine production and increased transcriptional activation of immune response genes in vaccinated (2 doses) vs unvaccinated individuals. Epigenetically, this study claims that vaccinations cause type of long-lasting memory at the level of partially sustained H3K27ac enrichment at particular gene promoters.

General remarks

The authors substantially improved the manuscript, primarily with new H3K27ac data stratification. The visualization and clarity of the content has also been enhanced. As mentioned in point-by-point response, major conclusions of the study have not changed through new re-analyses and were, rather, pointed toward more specific features (e.g. more pronounced H3K27ac retention at t4 at the promoters, compared to gene bodies and distal elements). The authors updated discussion and added missing information to method descriptions. An interesting enrichment with G4 quadruplexes in cluster 1 promoters have been added. Overall, streamlining of the manuscript and more optimal analyses were noticed. However, there are still issues in the analyses (or their interpretation) which are supposed to support major study claims about the persistence of H3K27ac-related epigenetic memory. Please see the points below.

Major points

H3K27ac data:

The authors substantially improved the quality of data visualization and descriptions of their methodology. They re-analyzed Cut&Run-seq with stratification to genomic elements as showed representative enrichment snapshots, as previously pointed out. I also acknowledge a change from superficial GO-centric stratification to unsupervised clustering. A key finding in the study is the long-term memory of stimulated macrophage responsiveness even in 3rd vaccination round, 6 months after previous administration. Importantly, such responsiveness was related to sustained H3K27ac remodelling. Unfortunately, despite substantial improvements in the paper, H3K27ac should be re-visited to support major claims of the study:

- Fig.3B: enrichment of H3K27ac in t4/t0 at the promoters is higher than 0 as the only feature showing such trend and implying an unique regulation. However no statistics are provided. Fig 3E provides comparison between genomic elements with statistics but they should be included for comparison between the timepoints to support the claim about long term retention of H3K27ac. The most crucial comparison would be in this case t4 vs t0 or t2/t0 vs t5/t0.
- Even though clustering properly depicts global H3K27ac remodelling across all timepoints and should be included as such, the claim about epigenetic memory and its relation to responsiveness or transcriptomic changes should be supported also by more specific comparison. That one would be: identify genes with H3K27ac-retained (significant increase t4 vs t0 or t5/t0 vs t2/t0) promoters and assess their transcriptomic changes (t2/t0) and gene ontology. The authors included similar analyses in Fig. EV2A,B and 5A, but they correspond to broad cluster 1 where similarity of the genes might come from changes at the other timepoints, not only the ones indicating memory (as seen in the t2-t3 timepoint difference between clusters in Response Fig. 1)
- The general enrichment of H3K27ac in the presented snapshots does imply a broad distribution, even in zoom in views and acknowledging log2 scale. However, H3K27ac is known to form more narrow peaks at promoters and cis regulatory elements (as mentioned by the authors too). In addition, t5/t0 shows increase over previous timepoints at a global, genome-wide scale, irrespectively of the gene or chromosomal region shown. Lowly expressed genes or genomic elements with overall low H3K27ac enrichment could show artificially amplified H3K27ac enrichment through division by low, noisy denominator (t0). The authors should show a supplementary data for H3K27ac changes (similar to Fig. 3B) but from the peaks non-normalized to t0 (only spike-in normalization and/or total read count (library size) normalization).

Gene expression data:

No further comments.

Discussion:

No further comments. The authors successfully added relevant discussion parts. The trigger that allows formation of G4 quadruplexes and their claimed stability is interesting to investigate but beyond the scope of the study.

Linking H3k27ac and transcription data:

The authors showed an enrichment of SP-stimulated genes (t2) in cluster 1 in new Fig. EV2B, as well as the trend of H3K27ac enrichment for SP-stimulated genes and cluster 1 in Fig. EV2A. However, more specific comparison will support study claims as described in the above points (t4 vs t0 or t5/t0 vs t2/t0 in H3K27ac data).

Minor points:

- Fig. EV4: comment on lack of difference for ssRNA in EV4 given the fact that it shows the highest retention of responsiveness in t3 vs t0 (Fig. 1C)

Point-by-point response for Simonis et al.: Persistent epigenetic memory of SARS-CoV-2 mRNA vaccination in monocyte-derived macrophages (MSB-2023-11820R)

We thank the reviewers for taking the time to review our revised manuscript. We have carefully addressed all the points raised by reviewer #3 in a detailed, point-by-point format. The reviewer comments are presented in gray and italics, while our responses are provided in blue font.

Reviewer #3:

All of my concerns have been addressed.

Response:

We sincerely thank Reviewer #3 for reviewing our manuscript and greatly appreciate the positive feedback.

Reviewer #4:**Summary**

Modified from summary in revision round 1. This study explores the transcriptional responses and associated chromatin changes in monocyte-derived macrophages of human subjects following two SARS-CoV-2 vaccination doses. Monocytes-derived macrophages isolated after 2 weeks following this regime showed increased responsiveness to in vitro stimulation, which remained high (significant only for ssRNA-stimulation) up to 10 weeks following second vaccination. High responsiveness is achieved after two vaccinations, suggesting prime-boost effect. The authors related changes in responsiveness with upregulated transcriptional activity of immune response genes, increased global enrichment of H3K27ac in the chromatin and enhanced cytokine production (all in stimulated macrophages from individuals with 2 vaccination doses (t2) vs unvaccinated (t0)). In addition, the authors showed that H3K27ac changes are long-lasting - its enrichment is further increased at the 3rd vaccination dose (t5; applied 6 months after the 2nd) and is retained long-term ~6 months after the 2nd dose (t4) specifically at gene promoters. The authors clustered H3K27ac data to identify gene group which belong to GOs for immune response genes, contain G4-quadruplexes in the promoters and show similar transcriptional profile to transcriptionally activated genes after 2nd vaccination (t2). Overall, this study suggests a long-term memory in monocyte-derived macrophages to SARS-CoV-2 vaccinations, correlated with increased responsiveness to immune triggers, increased stimulated cytokine production and increased transcriptional activation of immune response genes in vaccinated (2 doses) vs unvaccinated individuals. Epigenetically, this study claims

that vaccinations cause type of long-lasting memory at the level of partially sustained H3K27ac enrichment at particular gene promoters.

General remarks

The authors substantially improved the manuscript, primarily with new H3K27ac data stratification. The visualization and clarity of the content has also been enhanced. As mentioned in point-by-point response, major conclusions of the study have not changed through new re-analyses and were, rather, pointed toward more specific features (e.g. more pronounced H3K27ac retention at t4 at the promoters, compared to gene bodies and distal elements). The authors updated discussion and added missing information to method descriptions. An interesting enrichment with G4 quadruplexes in cluster 1 promoters have been added. Overall, streamlining of the manuscript and more optimal analyses were noticed. However, there are still issues in the analyses (or their interpretation) which are supposed to support major study claims about the persistence of H3K27ac-related epigenetic memory. Please see the points below.

Response:

We thank Reviewer #4 for taking the time to evaluate our manuscript and are pleased to receive positive feedback. Any remaining concerns have been addressed in a detailed point-by-point manner and in the revised manuscript.

Major points

Request 1:

H3K27ac data:

The authors substantially improved the quality of data visualization and descriptions of their methodology. They re-analyzed Cut&Run-seq with stratification to genomic elements as showed representative enrichment snapshots, as previously pointed out. I also acknowledge a change from superficial GO-centric stratification to unsupervised clustering. A key finding in the study is the long-term memory of stimulated macrophage responsiveness even in 3rd vaccination round, 6 months after previous administration. Importantly, such responsiveness was related to sustained H3K27ac remodelling. Unfortunately, despite substantial improvements in the paper, H3K27ac should be re-visited to support major claims of the study:

Fig.3B: enrichment of H3K27ac in t4/t0 at the promoters is higher than 0 as the only feature showing such trend and implying an unique regulation. However no statistics are provided.

Response 1:

We agree with the reviewer that the addition of further statistical analyses would improve our manuscript. It is worth noting that, due to the high number of data-points, all statistical comparisons between groups are highly significant ($P < 0.001$, calculated using One-Way ANOVA). However, in the context of comparing normalized H3K27ac coverage, quantifying the effect size is more meaningful than focusing solely on P-values. To address this, we have included a table with more detailed statistical information in both this point-by-point response and the manuscript to further support our observations (see Dataset EV6). In this context, we note that at t4 (normalized to t0), the median H3K27ac coverage at promoters is 5.74-fold higher compared to gene bodies and 7.43-fold higher compared to distal regions of the genes.

Promoters					
Distribution of H3K27ac coverage ratios (log2) across tx relative to t0					
Timepoint	t1/t0	t2/t0	t3/t0	t4/t0	t5/t0
Number	17026	17026	17026	17026	17026
Minimum	-8.272	-7.888	-8.216	-8.622	-9.86
25% Percentile	0.3461	1.615	0.9249	0.3257	1.743
Median	0.535	1.821	1.144	0.5984	1.981
75% Percentile	0.6818	1.982	1.403	0.7648	2.211
Maximum	7.768	7.841	10.38	6.97	9.321
Mean	0.4847	1.681	1.189	0.4498	1.977
Std. Deviation	0.6149	0.7157	0.8109	0.7479	0.8048
Std. Error of Mean	0.004712	0.005485	0.006214	0.005732	0.006168

Tukey's multiple comparisons test	Mean Diff.	Adjusted P Value
t1/t0 vs. t2/t0	-1.196	<0.0001
t1/t0 vs. t3/t0	-0.7044	<0.0001
t1/t0 vs. t4/t0	0.03495	<0.0001
t1/t0 vs. t5/t0	-1.492	<0.0001
t2/t0 vs. t3/t0	0.4917	<0.0001
t2/t0 vs. t4/t0	1.231	<0.0001
t2/t0 vs. t5/t0	-0.296	<0.0001
t3/t0 vs. t4/t0	0.7394	<0.0001
t3/t0 vs. t5/t0	-0.7877	<0.0001
t4/t0 vs. t5/t0	-1.527	<0.0001

Gene Bodies					
Distribution of H3K27ac coverage ratios (log2) across tx relative to t0					
Timepoint	t1/t0	t2/t0	t3/t0	t4/t0	t5/t0
Number	19583	19583	19583	19583	19583
Minimum	-9.036	-8.96	-8.183	-8.524	-8.232
25% Percentile	0.02845	0.7417	0.9225	-0.3337	1.682
Median	0.3653	1.243	1.379	0.1042	2.15
75% Percentile	0.689	1.63	1.851	0.4758	2.578
Maximum	7.168	7.68	7.647	7.002	9.399
Mean	0.35	1.134	1.369	0.02136	2.1
Std. Deviation	0.8057	0.908	1.086	0.9282	1.045
Std. Error of Mean	0.005758	0.006489	0.007758	0.006633	0.007469

Tukey's multiple comparisons test	Mean Diff,	Adjusted P Value
t1/t0 vs. t2/t0	-0.7838	<0.0001
t1/t0 vs. t3/t0	-1.019	<0.0001
t1/t0 vs. t4/t0	0.3287	<0.0001
t1/t0 vs. t5/t0	-1.75	<0.0001
t2/t0 vs. t3/t0	-0.2351	<0.0001
t2/t0 vs. t4/t0	1.112	<0.0001
t2/t0 vs. t5/t0	-0.9665	<0.0001
t3/t0 vs. t4/t0	1.348	<0.0001
t3/t0 vs. t5/t0	-0.7314	<0.0001
t4/t0 vs. t5/t0	-2.079	<0.0001

Distal regions					
Distribution of H3K27ac coverage ratios (log2) across tx relative to t0					
Timepoint	t1/t0	t2/t0	t3/t0	t4/t0	t5/t0
Number	19315	19315	19315	19315	19315
Minimum	-7.873	-7.755	-6.699	-8.573	-7.691
25% Percentile	-0.03288	0.5215	0.7918	-0.4352	1.535
Median	0.383	1.122	1.338	0.08051	2.094
75% Percentile	0.747	1.578	1.867	0.4895	2.575
Maximum	6.112	6.84	9.523	6.106	8.427
Mean	0.3477	0.9965	1.301	-0.02017	2.01
Std. Deviation	0.8412	0.9569	1.119	0.9693	1.103
Std. Error of Mean	0.006053	0.006885	0.008053	0.006974	0.007934

Tukey's multiple comparisons test	Mean Diff,	Adjusted P Value
t1/t0 vs. t2/t0	-0.6488	<0.0001
t1/t0 vs. t3/t0	-0.9531	<0.0001
t1/t0 vs. t4/t0	0.3679	<0.0001
t1/t0 vs. t5/t0	-1.662	<0.0001
t2/t0 vs. t3/t0	-0.3043	<0.0001
t2/t0 vs. t4/t0	1.017	<0.0001
t2/t0 vs. t5/t0	-1.013	<0.0001
t3/t0 vs. t4/t0	1.321	<0.0001
t3/t0 vs. t5/t0	-0.7089	<0.0001
t4/t0 vs. t5/t0	-2.03	<0.0001

Request 2)

Fig 3E provides comparison between genomic elements with statistics but they should be included for comparison between the timepoints to support the claim about long term retention of H3K27ac. The most crucial comparison would be in this case t4 vs t0 or t2/t0 vs t5/t0.

Response 2:

We thank the reviewer for pointing out a potential bias of the ratio analyses. We have now, similar to Fig 3B and as requested in 2) and 3), compared the distributions of the normalised K3K27ac read coverages (counts per million) in peaks located at promoters, gene bodies and distal regions at t4 vs t0 (new Fig. EV2A). In line with the ratio analyses, normalised read coverages are substantially and significantly increased at promoters for t4 vs t0 (median difference [MD] = 0.003, $P < 0.001$). In conclusion, both approaches, ratio (t4/t0) and the normalised coverage analysis (t4 vs t0) suggest long-term enrichment of H3K27ac at t4 relative to t0. We have included the additional analyses in Fig. EV2.

Request 3)

Even though clustering properly depicts global H3K27ac remodelling across all timepoints and should be included as such, the claim about epigenetic memory and its relation to responsiveness or transcriptomic changes should be supported also by more specific comparison. That one would be: identify genes with H3K27ac-retained (significant increase t_4 vs t_0 or t_5/t_0 vs t_2/t_0) promoters and assess their transcriptomic changes (t_2/t_0) and gene ontology. The authors included similar analyses in Fig. EV2A,B and 5A, but they correspond to broad cluster 1 where similarity of the genes might come from changes at the other timepoints, not only the ones indicating memory (as seen in the t_2 - t_3 timepoint difference between clusters in Response Fig. 1)

Response 3:

We apologize for the confusion, it appears that in our first rebuttal letter, data were not presented clear enough. We would like to clarify that the broader Cluster 1 exhibits a positive median ratio at t_4/t_0 , t_2/t_0 , and t_5/t_0 , indicating that these genes maintain persistent levels of H3K27ac across all critical time points. This figure was initially presented in the rebuttal letter only and is now shown in the manuscript (new Fig. EV2B). Notably, this cluster includes a specific subset of immune-related genes ($n = 333$) that we revealed through gene-ontology analysis. Importantly, these genes display increased H3K27ac ratios at their promoters at all key time points (t_4/t_0 , t_2/t_0 , and t_5/t_0) (new Fig. EV2C). The respective genes are also associated with enhanced expression in response to SP stimulation (new Fig. EV2D). Thus, our data conclusively demonstrate that cluster 1 and the immune-related genes within cluster 1 are significantly modified at the key time point t_4 versus t_0 .

Request 4)

The general enrichment of H3K27ac in the presented snapshots does imply a broad distribution, even in zoom in views and acknowledging \log_2 scale. However, H3K27ac is known to form more narrow peaks at promoters and cis regulatory elements (as mentioned by the authors too). In addition, t_5/t_0 shows increase over previous timepoints at a global, genome-wide scale, irrespectively of the gene or chromosomal region shown. Lowly expressed genes or genomic elements with overall low H3K27ac enrichment could show artificially amplified H3K27ac enrichment through division by low, noisy denominator (t_0). The authors should show a supplementary data for H3K27ac changes (similar to Fig. 3B) but from the peaks non-normalized to t_0 (only spike-in normalization and/or total read count (library size) normalization).

Response 4

We thank the reviewer for pointing out a potential bias of the ratio analyses. We have now, similar to Fig 3B and as requested in 2) and 3), compared the distributions of the normalised K3K27ac read coverages (counts per million) in peaks located at promoters, gene bodies and distal regions at t4 vs t0 (new Fig. EV2A). In line with the ratio analyses, normalised read coverages are substantially and significantly increased at promoters for t4 vs t0 (MD = 0.003, $p < 0.001$) in comparison to gene bodies (MD = 2.8×10^{-5} , $p < 0.001$) and distal regions (MD = 2.3×10^{-5} , $p < 0.001$) (Fig. 3B and Fig. EV2A). In conclusion, both approaches, ratio (t4/t0) and the normalised coverage analysis (t4 vs t0) suggest long-term enrichment of H3K27ac at t4 relative to t0. We have included the additional analyses in Fig. EV2.

Gene expression data:

No further comments.

Request 5)

Discussion:

No further comments. The authors successfully added relevant discussion parts. The trigger that allows formation of G4 quadruplexes and their claimed stability is interesting to investigate but beyond the scope of the study.

Response 5:

We thank the reviewer for his positive feedback regarding the additions to the discussion. We agree that investigating the trigger for G-quadruplex formation and their claimed stability is an intriguing direction for future research. However, as noted, this falls beyond the scope of the current study.

Request 6)

Linking H3k27ac and transcription data:

The authors showed an enrichment of SP-stimulated genes (t2) in cluster 1 in new Fig. EV2B, as well as the trend of H3K27ac enrichment for SP-stimulated genes and cluster 1 in Fig. EV2A. However, more specific comparison will support study claims as described in the above points (t4 vs t0 or t5/t0 vs t2/t0 in H3K27ac data).

Response 6:

We would like to refer to response 3 which covers the requested changes.

Request 7)

Minor points:

- Fig. EV4: comment on lack of difference for ssRNA in EV4 given the fact that it shows the highest retention of responsiveness in t3 vs t0 (Fig. 1C)

Response 7:

As correctly noted by the reviewer, the strongest retention of responsiveness at t3 versus t0 is observed for ssRNA. This sustained long-term responsiveness, coupled with increased levels of IL-1 β secreted after stimulation, may explain the relatively lower effects after a third vaccination compared to other stimuli, such as zymosan or Pam3Csk4, where a more pronounced decrease in IL-1 β secretion is observed over time after 2nd vaccination (e.g., at t3 (Figure 1C-E) or t4 (Figure 5D)). The reviewer's observation, along with a possible explanation, has now been incorporated into the revised manuscript (Lines 297–302).

24th Feb 2025

Manuscript number: MSB-2023-11820RR

Title: Persistent epigenetic memory of SARS-CoV-2 mRNA vaccination in monocyte-derived macrophages

Dear Jan,

Thank you again for sending us your revised manuscript. We are now satisfied with the modifications made and I am pleased to inform you that your paper has been accepted for publication.

Yours sincerely,
Jingyi

Jingyi Hou, PhD
Senior Editor
Molecular Systems Biology

Reviewer #4:

My concerns have been addressed. New figures and parts of text support authors' claims (e.g. Fig.EV2A & B). I recommend the manuscript for publication.
